



**Global methane budget and trend, 2010-2017: complementarity of inverse analyses**
**using in situ (GLOBALVIEWplus CH$_4$ ObsPack) and satellite (GOSAT) observations**
Xiao Lu[1], Daniel J. Jacob[1], Yuzhong Zhang[1,2,3], Joannes D. Maasakkers[4], Melissa P. Sulprizio[1], Lu Shen[1],
Zhen Qu[1], Tia R. Scarpelli[1], Hannah Nesser[1], Robert M. Yantosca[1], Jianxiong Sheng[5], Arlyn Andrews[6],
Robert J. Parker[7,8], Hartmut Boech[7,8], A. Anthony Bloom[9], Shuang Ma[9]
[1]John A. Paulson School of Engineering and Applied Sciences, Harvard University, Cambridge, MA,
USA
[2]School of Engineering, Westlake University, Hangzhou, Zhejiang Province, China
[3]Institute of Advanced Technology, Westlake Institute for Advanced Study, Hangzhou, Zhejiang Province,
China
[4]SRON Netherlands Institute for Space Research, Utrecht, The Netherlands.
[5]Center for Global Change Science, Massachusetts Institute of Technology, Cambridge, MA, USA
[6]National Oceanic and Atmospheric Administration, Earth System Research Laboratory, Boulder, CO,
USA
[7]National Centre for Earth Observation, University of Leicester, UK
[8]Earth Observation Science, Department of Physics and Astronomy, University of Leicester, UK
[9]Jet Propulsion Laboratory, California Institute of Technology, Pasadena, CA, USA
*Correspondence to:* Xiao Lu (xiaolu@g.harvard.edu) and Yuzhong Zhang
(zhangyuzhong@westlake.edu.cn)



**Abstract**

We use satellite (GOSAT) and in situ (GLOBALVIEWplus CH4 ObsPack) observations of atmospheric methane in a joint global inversion of methane sources, sinks, and trends for the 2010-2017 period. The inversion is done by analytical solution to the Bayesian optimization problem, yielding closed-form estimates of information content to assess the consistency and complementarity (or redundancy) of the satellite and in situ datasets. We find that GOSAT and in situ observations are to a large extent complementary, with GOSAT providing a stronger overall constraint on the global methane distributions, but in situ observations being more important for northern mid-latitudes and for relaxing global error correlations between methane emissions and the main methane sink (oxidation by OH radicals). The GOSAT observations achieve 212 independent pieces of information (DOFS) for quantifying mean 2010-2017 anthropogenic emissions on 1009 global model grid elements, and a DOFS of 122 for 2010-2017 emission trends. Adding the in situ data increases the DOFS by about 20-30%, to 262 and 161 respectively for mean emissions and trends. Our joint inversion finds that oil/gas emissions in the US and Canada are underestimated relative to the values reported by these countries to the United Nations Framework Convention on Climate Change (UNFCCC) and used here as prior estimates, while coal emissions in China are overestimated. Wetland emissions in North America are much lower than in the mean WetCHARTs inventory used as prior estimate. Oil/gas emissions in the US increase over the 2010-2017 period but decrease in Canada and Europe. Our joint GOSAT+in situ inversion yields a global methane emission of 551 Tg a$^{-1}$ averaged over 2010-2017 and a methane lifetime of 11.2 years against oxidation by tropospheric OH (86% of the methane sink).

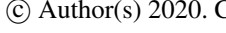



## 1 Introduction

Methane ($CH_4$) is the second most important anthropogenic greenhouse gas, and plays a central role in atmospheric chemistry as a precursor of tropospheric ozone and a sink of hydroxyl radicals (OH). It is emitted from many natural and anthropogenic sources that are difficult to quantify (Saunois et al., 2020). Atmospheric methane observations from satellites and in situ (surface, tower, shipboard, and aircraft) platforms have been used extensively to infer methane emissions and their trends through inverse analyses (Houweling et al., 2017). But the information from satellite and in situ observations does not always agree (Monteil et al., 2013; Bruhwiler et al., 2017) and is hard to compare because of large differences in observational density, precision, and the actual quantity being measured (Cressot et al., 2014). Here we use an analytical solution to the Bayesian inverse problem to quantitatively compare and combine the information from satellite (GOSAT) and in situ (GLOBALVIEWplus $CH_4$ ObsPack) observations for estimating global methane sources and their trends over the 2010-2017 period, including contributions from different source sectors and from the methane sink (oxidation by tropospheric OH).

Inverse analyses of atmospheric methane observations using chemical transport models (CTM) provide a formal method for inferring methane emissions and their trends (Brasseur and Jacob, 2017). Global satellite observations of atmospheric methane columns from the shortwave infrared SCIAMACHY and GOSAT instruments have been widely used for this purpose (Bergamaschi et al., 2013; Wecht et al., 2014; Turner et al., 2015; Maasakkers et al., 2019; Miller et al., 2019; Lunt et al., 2019). Other inverse analyses have relied on in situ methane observations that have much higher precision, are more sensitive to surface emissions, and include isotopic information, but are much sparser (Pison et al, 2009; Bousquet et al., 2011; Miller et al., 2013; Patra et al., 2016; McNorton et al., 2018).

A number of inverse analyses have combined in situ and satellite observations (Bergamaschi et al., 2007, 2009, 2013; Fraser et al., 2013; Monteil et al, 2013; Cressot et al., 2014; Houweling et al., 2014; Alexe et al., 2015; Ganesan et al., 2017; Janardanan et al., 2020), but few of them have compared the information from the two data streams and then mostly qualitatively. Bergamaschi et al. (2009, 2013), Fraser et al. (2014), and Alexe et al. (2015) found that surface and satellite methane observations provided consistent constraints on global methane emissions, but that satellite observations achieved stronger regional constraints in the tropics. No study to our knowledge has compared the ability of satellite and in situ observations to attribute long-term methane trends.

Analytical solution to the inverse problem, as used here, provides closed-form error characterization as part of the solution, and from there allows derivation of the information content from different components of the observing system (Rodgers, 2000). Application to satellite observations has been used to determine where the observations can actually constrain the inverse solution (Turner et al., 2015). The major obstacle to this analytical solution in the past has been the need to construct the Jacobian matrix for the CTM forward model, but this is now readily done using massively parallel computing clusters (Maasakkers et al., 2019). Such a method provides a means to quantify the differences in information





content between different data streams (e.g., satellite vs. in situ) and from there to contribute to the design
of a better observing system.

Here we apply satellite observations of atmospheric methane columns from the GOSAT instrument
together with an extensive global compilation of in situ observations (including surface, tower, shipboard,
and aircraft methane measurements) from the GLOBALVIEWplus $CH_4$ ObsPack v1.0 data product
(Cooperative Global Atmospheric Data Integration Project, 2019), to quantify the global distribution of
methane emissions, loss from reaction with OH, and related trends for the 2010-2017 period. We use for
this purpose an analytical inversion method that formally characterizes the information content from the
two data streams, whether that information is consistent, and whether it is complementary or redundant
(Rodgers, 2000; Jacob et al., 2016). Our work provides a comprehensive global perspective on the sources
contributing to 2010-2017 methane emissions and trends, as well as a general framework for synthesizing
the information from satellite and in situ observations.

**2 Methods**
Figure 1 summarizes the components of our analytical inversion system, which builds on previous
inversions of GOSAT satellite data by Maasakkers et al. (2019) and Zhang et al. (2019) but adds the in
situ observations. We apply observations $y$ from GLOBALVIEWplus observations and/or GOSAT
(Section 2.1), with the GEOS-Chem CTM as forward model (Section 2.3), to optimize the state vector $x$
of our inverse problem. The state vector has dimension $n = 3378$ including mean 2010-2017 non-wetland
methane emissions on the GEOS-Chem $4° \times 5°$ global grid ($n_1 = 1009$), 2010-2017 linear trends for these
emissions on that grid ($n_2 = 1009$), monthly mean wetland methane emissions for individual years in 14
subcontinental regions ($n_3 = 12 \times 8 \times 14 = 1344$), and tropospheric OH concentrations in each hemisphere
for individual years ($n_4 = 2 \times 8 = 16$). Section 2.2 describes the prior state vector estimates ($x_A$) and the
prior error covariance matrix ($S_A$). We derive posterior estimates $\hat{x}$ of the state vector and the associated
error covariance matrix $\hat{S}$ by analytical solution to the Bayesian optimization problem (Section 2.4). We
present results from three inversions using in situ observations only (In situ-only inversion), GOSAT
observations only (GOSAT-only inversion), and both GOSAT and in situ observations (GOSAT + in situ
inversion).

**2.1 Methane observations**
The GLOBALVIEWplus $CH_4$ ObsPack v1.0 data product compiled by the National Oceanic and
Atmospheric Administration (NOAA) Global Monitoring Laboratory includes worldwide high-accuracy
measurements of atmospheric methane concentrations from different observational platforms (surface,
tower, shipboard, and aircraft) (Cooperative Global Atmospheric Data Integration Project, 2019). Here
we use the ensemble of GLOBALVIEWplus observations for 2010-2017. For surface and tower
measurements, we use only daytime (10-16 local time) observations and average them to the
corresponding daytime mean values. We exclude outliers at individual sites that depart by more than three
standard deviations from the mean. We obtain in this manner 157054 observation data points for the



inversion including 81119 from 103 surface sites, 27433 from 13 towers, 827 from 3 ship cruises, and
47675 from 29 aircraft campaigns. Figure 2a shows the mean methane concentrations in 2010-2017 from
the in situ data. The data are relatively dense in North America and western Europe, with also a few sites
in China, but otherwise mainly measure background concentrations. The number of available surface and
tower observations increases from 10493 in 2010 to 19657 in 2017 with largest changes in Europe and
Canada.
GOSAT is a nadir-viewing instrument in space since 2009 that measures the backscattered solar radiation
from a sun-synchronous orbit at around 13:00 local time (Butz et al., 2011; Kuze et al, 2016). Observing
pixels are 10-km in diameter and separated by about 250 km along-track and cross-track in normal
observation mode, with higher-density data collected in targeted observation modes. Methane is retrieved
at the 1.65 µm absorption band. We use dry column methane mixing ratios from the University of
Leicester version 9.0 Proxy $XCH_4$ retrieval (Parker et al., 2020). The retrieval has a single-observation
precision of 13 ppb and a regional bias of 2 ppb (Buchwitz et al., 2015). We use GOSAT data for 2010-
2017 including 1.6 million retrievals over land as shown in Figure 2b. We do not use glint data over the
oceans and data poleward of 60ºN because of seasonal bias and the potential for large errors (Maasakkers
et al., 2019).
**2.2 Prior estimates**
Table1 summarizes the prior estimates of the mean 2010-2017 methane emissions used for the state vector,
and Figure 3 shows the spatial patterns. Natural sources include the ensemble mean of the WetCHARTS
inventory version 1.2.1 (Bloom et al., 2017) for wetlands, open fires from the Global Fire Emissions
Database version 4s with seasonal and interannual variability (van der Werf et al., 2017), termites from
Fung et al. (1991), and seeps from Etiope et al. (2019) with global scaling to 2 Tg a$^{-1}$ from Hmiel et al.
(2020). The default anthropogenic emissions are from EDGAR v4.3.2 (Janssens-Maenhout et al., 2019),
and are superseded for fugitive fuel emissions (oil, gas, coal) by the Scarpelli et al. (2020) inventory
which spatially allocates national emissions reported by countries to the United Nations Framework
Convention of Climate Change (UNFCCC). US anthropogenic emissions are further superseded by the
gridded version of Inventory of U.S. Greenhouse Gas Emissions and Sinks from the Environmental
Protection Agency (EPA GHGI) (Maasakkers et al., 2016). The WetCHARTS wetlands inventory includes
seasonal and interannual variability that is optimized in the inversion through correction to the monthly
emissions. Seasonality from Zhang et al. (2016) is imposed for rice emissions, and temperature-dependent
seasonality is applied to manure emissions (Maasakkers et al., 2016). Other emissions are aseasonal.
We assume a 50% error standard deviation for all anthropogenic and non-wetland natural emissions on
the 4º latitude ×5º longitude grid, with no spatial error covariance so that their prior error covariance
matrix is diagonal, which is a reasonable assumption for anthropogenic emissions (Maasakkers et al.,
2016). We assume 0 ± 10% a$^{-1}$ as prior estimate for the linear 2010-2017 emission trends on the 4º×5º
grid. Prior estimates of monthly mean wetland methane emissions for individual years in 14





subcontinental regions, along with their error covariance matrix, are from the WetCHART v1.2.1
inventory ensemble (Bloom et al., 2017). The prior methane emissions total 533 Tg a$^{-1}$, at the low end of
the current top-down estimates (538-593 Tg a$^{-1}$) for 2008-2017 (Saunois et al., 2020), and this largely
reflects the downward revision of global seep emissions by Hmiel et al. (2020).

Prior monthly 3-D fields of global tropospheric OH concentrations are taken from a GEOS-Chem
simulation with full-chemistry (Wecht et al., 2014) that yields a methane lifetime $\tau_{CH_4}^{OH}$ due to oxidation
by tropospheric OH of 10.6±1.1 years and an inter-hemispheric OH ratio (North to South) of 1.16. The
methane lifetime is consistent with the value of 11.2±1.3 years inferred from methylchloroform
observations (Prather et al., 2012), while the inter-hemispheric OH ratio is slightly higher than the
observed range of 0.97±0.12 (Patra et al., 2014) but closed to recent multi-model estimates of 1.3±0.1
(Zhao et al., 2019). We assume no interannual variability in this prior OH field. We use 10% as prior error
standard deviation for the hemispheric OH concentrations in individual years, based on Holmes et al.
(2013). Corrections to OH in the inversion are applied as a hemispheric scaling factor for individual years,
without changing the spatial or temporal pattern of the original fields. Zhang et al. (2018) conducted
methane inversions with twelve different OH fields from the ACCMIP model ensemble (Naik et al., 2013)
and found no significant difference with the GEOS-Chem OH fields used here except for two outlier
models.

**2.3 Forward Model**
We use the GEOS-Chem 12.5.0 (http://geos-chem.org) global CTM (Bey et al., 2001; Wecht et al., 2014;
Maasakkers et al., 2019) as forward model to simulate atmospheric methane concentrations and their
sensitivity to the state vector elements. The model is driven by MERRA-2 reanalysis meteorological fields
from the NASA Global Modeling and Assimilation Office (GMAO) (Gelaro et al., 2017). The methane
sink is computed within the model from 3-D tropospheric oxidant fields including OH (optimized in the
inversion), Cl atoms (Wang et al., 2019), 2-D stratospheric oxidant fields (Murray et al., 2012), and soil
uptake (Murguia-Flores et al., 2018). We conduct GEOS-Chem model simulations for 2010-2017 at
global 4°× 5° resolution with 47 vertical layers extending to the mesosphere. GEOS-Chem has excessive
methane in the high-latitudes stratosphere, a flaw common to many models (Patra et al., 2011), and we
correct for this bias, with stratospheric methane profiles measured by the solar occultation ACE-FTS v3.6
instrument (Waymark et al., 2014; Koo et al., 2017) following Zhang et al. (2019). Initial model conditions
on January 1, 2010 are set to be unbiased in zonal mean relative to GOSAT observations for January 2010,
and we find that they are also unbiased relative to the GLOBALVIEWplus in situ observations. In this
manner, model discrepancies with observations over the 2010-2017 period can be attributed to model
errors in emissions or OH over that period, instead of error in initial conditions. We archive model
methane dry mixing ratios at each location and time of the in situ and GOSAT datasets for 2010-2017.

As forward model $\boldsymbol{F}$ for the inversion, GEOS-Chem relates the state vector $\boldsymbol{x}$ to the atmospheric
concentrations $\boldsymbol{y}$ as $\boldsymbol{y} = \boldsymbol{F}(\boldsymbol{x})$ (Fig.1). The simulation of observations with the prior estimates of state
vectors ($x_A$) in 2010-2017 diagnoses systematic errors in comparison to observations that enable
improved estimate of the state vector through the inversion. In addition, the random component of the
discrepancy can be used to estimate the observation error (sum of instrument error, representation error,
and forward model error) in the Bayesian optimization problem using the residual error method (Heald et
al., 2004). The method assumes that the systematic component of the model bias $(\overline{y - F(x_A)})$ for
individual years, where the overbar denotes the temporal average in a 4°× 5° grid cell (for GOSAT) or for
an observation platform (for in situ observations), is to be corrected in the inversion, while the residual
term ($\varepsilon_0 = y - F(x_A) - \overline{y - F(x_A)}$) represents the random observation error. Here we applied this
method to construct the observation error covariance matrix $S_o$ from the statistics of $\varepsilon_0$.

We find that the mean standard deviation of the random observation error ($\varepsilon_0$) for the GLOBALVIEWplus
in situ data averages 36 ppbv (20 and 45 ppbv for background and non-background surface observations,
68 ppbv for tower observations, 10 ppbv for shipboard observations, 24 ppbv for aircraft observations),
compared to 13 ppbv for GOSAT. The observation error for in situ observations is dominated by the
forward model error while for GOSAT it is dominated by the instrument error. The forward model error
is higher for surface concentrations near source regions than for columns or other in situ observations
measuring background, because the amplitude of methane variability is much higher (Cusworth et al.,
2018) and more challenged for a model at 4°× 5° resolution to capture. We assume that $S_o$ is diagonal
in the absence of better objective information, but in fact some error correlation between different
observations could be expected to arise from transport and source aggregation errors in the forward model.
This is considered by introducing a regularization factor $\gamma$ in the minimization of the cost function for
the inversion (Section 2.4).

**2.4 Analytical Inversion**
Bayesian solution to the state vector optimization problem assuming Gaussian prior and observation
errors involves minimizing the cost function $J(x)$:
$$J(x) = (x - x_A)^T S_A^{-1} (x - x_A) + \gamma (y - F(x))^T S_O^{-1} (y - F(x)) \quad (1),$$
where $x$ is the state vector, $x_A$ denotes the prior estimate of $x$, $S_A$ is the prior error covariance matrix,
$y$ is the observation vector, $F(x)$ represents the GEOS-Chem simulation of $y$, $S_O$ is the observation
error covariance matrix, and $\gamma$ is a regularization factor. The need for $\gamma$ in $J(x)$ is to avoid giving
excessive weighting to observations, due to the likely underestimate of $S_O$ when unknown error
correlations are not included in its construction (Zhang et al., 2018; Maasakkers et al., 2019). γ here plays
the same role as the regularization parameter in Tikhonov methods (Brasseur and Jacob, 2017) and reflects
our inability to properly quantify the magnitude of errors.

Minimization of the cost function in equation (1) has an analytical solution if the forward model is linear
(Rodgers, 2000). The inverse problem here is not strictly linear regarding the optimization of OH, because
the sensitivity of the methane concentration to change in OH concentrations depends on the methane
concentration through first-order loss. The variability of methane concentration is sufficiently small that





this non-linearity is negligible. We thus express the GEOS-Chem forward model as $\boldsymbol{y} = \boldsymbol{Kx} + \boldsymbol{c}$, where
$\boldsymbol{K} = \partial \boldsymbol{y}/\partial \boldsymbol{x}$ represents the Jacobian matrix and $\boldsymbol{c}$ is an initialization constant. We construct the Jacobian
matrix $\boldsymbol{K}$ explicitly by conducting GEOS-Chem simulations with each element of the state vector
perturbed separately. Minimizing the Bayesian cost function by solving $dJ(\boldsymbol{x})/d\boldsymbol{x} = \boldsymbol{0}$ yields closed-
form expressions for the posterior estimate of the state vector $\widehat{\boldsymbol{x}}$ and its error covariance matrix $\widehat{\boldsymbol{S}}$:
$$\widehat{\boldsymbol{x}} = \boldsymbol{x}_A + \boldsymbol{G}(\boldsymbol{y} - \boldsymbol{Kx}_A) \quad (2),$$

$$\widehat{\boldsymbol{S}} = (\gamma \boldsymbol{K}^T \boldsymbol{S}_O^{-1} \boldsymbol{K} + \boldsymbol{S}_A^{-1})^{-1} \quad (3),$$


where $\boldsymbol{G}$ is the gain matrix,
$$\boldsymbol{G} = \frac{\partial \widehat{\boldsymbol{x}}}{\partial \boldsymbol{y}} = (\gamma \boldsymbol{K}^T \boldsymbol{S}_O^{-1} \boldsymbol{K} + \boldsymbol{S}_A^{-1})^{-1} \gamma \boldsymbol{K}^T \boldsymbol{S}_O^{-1} \quad (4).$$


From the posterior error covariance matrix one can derive the averaging kernel matrix describing the
sensitivity of the posterior estimate to the true state:
$$\boldsymbol{A} = \frac{\partial \widehat{\boldsymbol{x}}}{\partial \boldsymbol{x}} = \boldsymbol{I}_n - \widehat{\boldsymbol{S}} \boldsymbol{S}_A^{-1} \quad (5).$$

The trace of $\mathbf{A}$ quantifies the degrees of freedom for signal (DOFS), which represents the number of
pieces of independent information gained from the observing system for constraining the state vector
(Rodgers, 2000).

We choose the value for the regularization parameter $\gamma$ in order to achieve a solution most consistent
with the estimated error on the prior estimates. For a given state vector element $i$, the expected value of
$(\widehat{\boldsymbol{x}}_i - \boldsymbol{x}_{Ai})^2$ is the prior error variance $\sigma_{Ai}^2$. For a diagonal prior error covariance matrix, the state
component $J_A$ of the posterior cost function is
$$J_A(\widehat{\boldsymbol{x}}) = (\widehat{\boldsymbol{x}} - \boldsymbol{x}_A)^T \boldsymbol{S}_A^{-1}(\widehat{\boldsymbol{x}} - \boldsymbol{x}_A) = \sum_n \frac{(\widehat{x}_i - x_{Ai})^2}{\sigma_{Ai}^2} \approx n \quad (6),$$


where $n$ is the number of state vector elements. In our case the prior error covariance matrix is not strictly
diagonal because of covariance for the wetland terms (Bloom et al., 2017), so one may expect $J_A(\widehat{\boldsymbol{x}})$ to
be somewhat deviated from $n$. Nevertheless, $J_A(\widehat{\boldsymbol{x}}) \gg n$ implies overfit to the observations because the
posterior state vector estimates are far outside the estimated errors on the prior estimates.

One can apply the same reasoning to the observation component $J_O$ of the posterior cost function,
$$J_O(\widehat{\boldsymbol{x}}) = (\boldsymbol{y} - \boldsymbol{K}\widehat{\boldsymbol{x}})^T \boldsymbol{S}_O^{-1}(\boldsymbol{y} - \boldsymbol{K}\widehat{\boldsymbol{x}}) \approx m \quad (7),$$


where $m$ is the number of observations. However, this component is less sensitive to the choice of $\gamma$
because of the large random error component for individual observations.

Figure 4 shows the dependences of $J_A(\widehat{\boldsymbol{x}})$ and $J_O(\widehat{\boldsymbol{x}})$ on the choice of the regularization parameter $\gamma$, for
the in situ and GOSAT observations. The in situ observations are sufficiently sparse that $\gamma = 1$ (no





regularization) provides the best solution. In the case of GOSAT, however, $\gamma = 1$ would yield $J_A(\hat{x}) = 6n$
which indicates overfit, while $\gamma = 0.1$ yields $J_A(\hat{x}) \approx n$ which is the expected value and is used here.
This can be explained by the high observation density of GOSAT, such that error correlation between
individual observations may be expected and would have a large effect on the solution. Maasakkers et al.
(2019) found that $\gamma = 0.05$ and $\gamma = 0.1$ gave similar solutions in their global inversions of GOSAT
data.

Analytical solution to the cost function minimization problem, as done here, has several advantages
relative to the more commonly used variational (numerical) approach for finding the minimum. (1) It
finds the true minimum in the cost function, rather than an approximation that may be sensitive to the
choice of initial estimate. (2) It identifies the information content of the inversion and the ability to
constrain each state vector element. (3) It enables a range of sensitivity analyses modifying the prior
estimates, modifying the error covariance matrices, adding/subtracting observations, etc. at minimal
computational cost. We will make use of these advantages in comparing the ability of the In situ-only,
GOSAT-only, and GOSAT + in situ inversions. A requirement of the analytical approach is that the
Jacobian matrix be explicitly constructed, requiring $n + 1$ forward model runs. However, this construction
is readily done in parallel on high-performance computing clusters.

Our inversion returns posterior emission estimates and their temporal trends on a $4° \times 5°$ grid for non-
wetland emissions, and monthly mean wetland emissions for individual years in 14 subcontinental regions.
We can aggregate these results spatially and by sector in a way that retains the error covariance of the
solution (Maasakkers et al., 2019). Consider a reduced state vector $x_{red}$ representing a linear combination
of the original state vector elements that may be a sum over a particular region or the globe, and may be
weighted by the contributions from individual sectors following the prior distribution. The linear
transformation from the posterior full-dimension state vector $\hat{x}$ to the reduced state vector $\hat{x}_{red}$ is
defined by a summation matrix $W$

$$\hat{x}_{red} = W\hat{x} \quad (8).$$


The posterior error covariance and averaging kernel matrices for the reduced state vector can then be
calculated as:

$$\hat{S}_{red} = W\hat{S}W^T \quad (9),$$
$$A_{red} = WAW^* \quad (10),$$

where $W^* = W^T(W\,W^T)^{-1}$ (Calisesi et al., 2005). $\hat{S}_{red}$ provides a means to determine error
correlations between aggregates of quantities optimized by the inversion, e.g., between global methane
emissions and global OH concentrations. $A_{red}$ provides a means to determine the ability of the inversion
to constrain an aggregated term (e.g., emissions from a particular sector).

**3. Results and discussion**
**3.1 Ability to fit the in situ and GOSAT data**





We will present results from three different inversions for 2010-2017: (1) using only in situ observations
(In situ-only inversion), (2) using only GOSAT observations (GOSAT-only inversion), and (3) using both
GOSAT and in situ observations (GOSAT + in situ inversion). Here we first evaluate the ability of these
different inversions to fit the in situ and GOSAT observations, including when the data are not used in the
inversion (consistency check). This is done by conducting GEOS-Chem simulations with posterior values
for the state vectors and comparing to observations.

Figures 5 and 6 show the resulting comparisons for the in situ observations, arranged by type of platform
(Fig.5), and by latitude bands and months (panels (a)-(d) in Fig.6). The model simulation with prior
estimates shows a 30-60 ppb low bias for all in situ platforms growing with time. The In situ-only
inversion effectively corrects this bias and its trend, and also significantly improves the correlations across
all platforms. The GOSAT-only inversion performs comparably in correcting the bias for the independent
aircraft data measuring the background, and also corrects the 2010-2017 trend, but still shows notable
low bias at northern mid-latitudes because of difficulty in fitting the surface and tower data in the US and
Europe that are adjacent to methane sources.

Figure 6 also compares the fits to the GOSAT observations (panels (e)-(h)). Both the In situ-only and
GOSAT-only inversions correct the bias and trend in the prior simulation at all latitudes. An important
implication is that the in situ observations, even though sparse and mostly at northern mid-latitudes, can
still inform the global methane levels. The GOSAT + in situ joint inversion shows good agreement with
both the in situ and GOSAT observations.

Figure 7 further evaluates the global methane growth rate as determined by the methane budget imbalance
for individual year in 2010-2017 from the three inversions. The observed methane growth rate inferred
from the NOAA sites (https://www.esrl.noaa.gov/gmd/ccgg/trends_ch4/, last access: 20 June 2020)
averages $7.2\pm2.8$ ppb $a^{-1}$ over the period, peaking in 2014, and overall accelerating with higher growth in
2015-2017 than in 2010-2013. We find that all posterior simulations show comparable mean methane
growth rate ($7.7\pm3.7$ ppb $a^{-1}$ for In situ-only inversion, $8.8\pm2.2$ ppb $a^{-1}$ for GOSAT-only inversion, and
$8.3\pm1.8$ ppb $a^{-1}$ for the GOSAT + in situ inversion). However, the In situ-only inversion overestimates the
increasing trend in the methane growth rate, largely driven by year 2017, and fails to fit its interannual
variability. This may reflect the heavy weighting of the in situ observations toward northern mid-latitudes.
GOSAT observations in the inversion do much better in capturing the observed methane interannual
variability and trend. Adding in situ observations to GOSAT observations provides a better fit in 2015
than GOSAT-only inversion but has insignificant effect in other years. Zhang et al. (2019) interpreted the
trend and interannual variability in the GOSAT-only inversion as due to a combination of anthropogenic
emissions, wetlands, and OH concentrations.

**3.2 Anthropogenic methane emissions**

Figure 8 shows the averaging kernel sensitivities (diagonal elements of the averaging kernel matrix) and



posterior scaling factors for the non-wetland emissions (dominated by anthropogenic emissions) in the In
situ-only, GOSAT-only, and GOSAT + in situ joint inversions. The DOFS (trace of the averaging kernel
matrix) quantify the number of independent pieces of information from the inversion, starting from 1009
unknowns for anthropogenic emissions (Figure 1). The DOFS are 113 for the In situ-only inversion, 212
for the GOSAT-only inversion, and 262 for the GOSAT + in situ joint inversion. The higher DOFS from
the joint inversion indicate that the satellite and in situ observations have complementarity but also some
redundancy. Strict complementarity would imply a DOFS of 325=113+212. We find that 75% of the in
situ information is at northern mid-latitudes (30-60°N, DOFS=82, calculated as the sum of averaging
kernel sensitivities in that latitude band) where the observations are densest, with another 9% (DOFS=10)
at 60-90°N. GOSAT provides higher information than in situ observations at northern mid-latitudes
(DOFS=96) and dominates in the tropics (DOFS=105). This dominance of satellites for informing
methane sources in the tropics has been pointed out in previous studies (Bergamaschi et al., 2013; Monteil
et al., 2013; Fraser et al., 2013; Alexe et al., 2015).
We investigate further the inversion results for northern mid-latitudes where most of the information of
in situ observations is contained including for the US, Canada, Europe, and China. Table 2 gives the
optimization of anthropogenic methane emissions (calculated as the difference between total non-wetland
emissions and the non-wetland natural emissions) in these regions. Figure 9 shows the optimization by
source sectors, assuming that (1) the partitioning between sectors of non-wetland emissions in individual
grid cells is correct in the prior inventory (this does not assume that the prior distribution of sectoral
emissions is correct)., (2) the scaling factors are to be applied equally to all sectors in a grid cell. These
assumptions are adequate when the sectors are spatially separated but are more prone to error when they
spatially overlap. Figure 9 also shows the averaging kernel sensitivities of emission sectors (diagonal
terms of $A_{red}$ derived from Equations (8) and (10)), measuring the ability of the inversion to optimize
different emissions sectors, and the DOFS for each inversion summed over the region. Wetland methane
emissions are optimized separately as will be discussed in Section 3.3.
Inspection of the DOFS shows that the in situ observations are more effective than GOSAT for optimizing
US anthropogenic methane emissions (DOFS=41 vs. DOFS=22) and this applies to all sectors (Figure 9).
The averaging kernel sensitivities panel in Figure 9 shows that US results from the joint GOSAT + in situ
inversion are mostly determined by the in situ observations. The joint GOSAT + in situ inversion increases
anthropogenic US emissions from 28 Tg a$^{-1}$ in the prior EPA GHGI to 36 Tg a$^{-1}$, with most of the increase
driven by livestock and oil/gas sources in the central US. Averaging kernel sensitivity for major sectors
is large (0.63-0.93), indicating that the posterior estimates are mostly determined by the observations
rather than by the prior estimates. The underestimate of oil/gas emissions in the EPA GHGI has been
reported before (Alvarez et al., 2018; Cui et al., 2019; Maasakkers et al., 2020).
In situ observation is also more effective than GOSAT in optimizing anthropogenic methane emissions in
Canada (DOFS=21 vs. DOFS=6), particularly in Alberta where oil/gas emissions are high (Fig.8). This





reflects in part our exclusion of GOSAT data poleward of 60°N. Oil/gas emissions in Canada increase by
a factor of 2 in the GOSAT + in situ inversion to 4.5 Tg a$^{-1}$ compared to the ICF (2015) prior estimate,
with an averaging kernel sensitivity of 0.57 (Fig.9). Total anthropogenic emissions increase from 5 Tg a$^{-}$
$^{1}$ to 8 Tg a$^{-1}$.

In situ and GOSAT observations show comparable ability in optimizing the total anthropogenic emissions
in Europe (DOFS=16~18). They agree that prior anthropogenic methane emissions are too high in
northern Europe but disagree in southern Europe. Averaging kernel sensitivities from the In situ-only
inversion are slightly weaker than for the US and Canada because of the lower density of in situ sites. The
Integrated Carbon Observation system (ICOS) network (https://www.icos-cp.eu/, last access: 17 July
2020) has increased substantially the number of available methane observations in Europe since 2017 so
that future inversions should expect a stronger constraint from in situ observations. Total European
anthropogenic emissions decrease from 27 Tg a$^{-1}$ to 23 Tg a$^{-1}$ in the GOSAT + in situ joint inversion, with
decreases for all sectors but this may reflect the inability of our 4°× 5° resolution to effectively separate
emission sectors.

The only other region where in situ observation provides significant information is China, though the
corresponding DOFS=13 is less than for GOSAT (DOFS=22). Both inversions agree that emissions must
be greatly decreased from the prior estimate, and the joint inversion (DOFS=28) has stronger power in
doing so. The posterior 2010-2017 Chinese anthropogenic emission is 43 Tg a$^{-1}$ in the joint inversion,
compared to 63 Tg a$^{-1}$ in the prior estimate. Our results agree with a recent study by Janardanan et al.
(2020), which also used GOSAT and surface observations to estimate a mean 2011-2017 anthropogenic
methane emission in China of 46±9 Tg a$^{-1}$. The downward correction is mainly driven by a 40% decrease
in coal emissions from 19 Tg a$^{-1}$ to 11 Tg a$^{-1}$ (Fig. 9). Previous inversions using the EDGAR inventory
(>20 Tg a$^{-1}$) as prior estimate found a similar correction (Alexe et al., 2015; Thompton et al., 2015; Turner
et al., 2015; Maasakkers et al., 2019; Miller et al., 2019). In our case, the prior estimate of coal emissions
(19 Tg a$^{-1}$) is the value reported by China to the UNFCCC and we find that it is still too high. A recent
inventory by Sheng et al. (2019) gives a coal emission estimate of 15 Tg a$^{-1}$ for China in 2010-2016.

**3.3 Wetland methane emissions**
The inversion optimizes wetland emissions for the 14 regions of Figure 3 and for 96 individual months
covering 2010-2017, amounting to 1344 state vector elements. Results from the In situ-only, GOSAT-
only, and GOSAT + in situ inversions yield DOFS of 221, 183, and 301 respectively. In situ observations
provide more information for boreal wetlands while GOSAT dominates for tropical wetlands.

We analyzed further the boreal/temperate North America wetlands, where in situ observations provide
significant added information (Figure 10). Both in situ and GOSAT observations agree that the prior
WetCHARTs emissions are too high. The posterior estimates from the GOSAT + in situ inversion are 4.5
and 2.0 Tg a$^{-1}$ for boreal and temperate North America, respectively, compared to 12.8 and 6.9 Tg a$^{-1}$ in





WetCHARTs. Posterior boreal wetland CH₄ emissions are on the lower end but within the WetCHARTs
estimates (WetCHARTs models range 3~33 Tg a⁻¹); however, posterior temperate CH₄ emissions are
lower and outside WetCHARTs range (3~12 Tg a⁻¹). The correction for boreal North America is
particularly large in May-June, which can potentially be attributed to suppression of wetland emissions
by either snow cover (Pickett-Heaps et al., 2011) or by frozen soils (Zona et al., 2016). The WetCHARTs
emission overestimate for temperate North America (mainly coastal wetlands in the eastern US) has been
reported before from inversions using aircraft data (Sheng et al., 2018) and GOSAT data (Maasakkers et
al., 2020).

### 3.4 Anthropogenic methane emission trends

Figure 11 presents the 2010-2017 trends of anthropogenic methane emissions from the three inversions,
and the corresponding averaging kernel sensitivities. The GOSAT + in situ inversion has a DOFS = 161
for quantifying the spatial distribution of the trends. Most of that information is from GOSAT (DOFS =
122) but in situ observations add significant information. Information from in situ observations is
concentrated in the US, Canada, Europe, and China. Table 2 summarizes the trends for the four regions.
Figure 12 shows the trends disaggregated by sectors, using the same procedure as for Figure 9.
In situ observations provide stronger constraints than GOSAT on anthropogenic emission trends in the
US (DOFS=29 vs. DOFS=12). They agree on the upward trend in the eastern US as also found in
Maasakkers et al. (2020) which used GOSAT in a high resolution inversion to interpret methane trends in
US in 2010-2015. However, they show opposite trends (positive trend from In situ-only inversion but
negative from GOSAT-only inversion) in total emissions and in the central south US (Table 2, Fig. 11).
The GOSAT + in situ joint inversion (DOFS=31) estimates that US anthropogenic methane emissions
increased by 0.4 Tg a⁻¹ a⁻¹ (1.1% a⁻¹) from 2010 to 2017, with the largest contribution from oil/gas
emissions (0.3 Tg a⁻¹ a⁻¹, 2.5% a⁻¹). This posterior trend is much smaller than previous studies showing
large increases in US oil/gas emissions (2.1–4.4 Tg a⁻¹ a⁻¹) inferred from ethane/propane levels (Franco
et al., 2016; Hausmann et al., 2016; Helmig et al., 2016), but is more consistent with a recent study by
Lan et al. (2019) of 0.3±0.1 Tg a⁻¹ a⁻¹ in 2006-2015 based on long-term in situ measurements. The
inversion also reveals rising emissions from oil/gas in the central south US, including the Permian Basin
which is currently the largest oil-producing basin in the US (Zhang et al., 2020).
We find that anthropogenic emissions in Canada decrease over the 2010-2017 period by 0.2 Tg a⁻¹ a⁻¹
(2.5% a⁻¹) in the GOSAT + in situ joint inversion, mostly driven by oil/gas emissions in Alberta and
livestock emissions (Figs. 11-12). Anthropogenic emissions in Europe decrease by 0.4 Tg a⁻¹ a⁻¹ (1.7 % a⁻
¹).
All three inversions show increases of Chinese anthropogenic methane emissions over 2010-2017 by 0.1-
0.4 Tg a⁻¹ a⁻¹ (0.3-0.9% a⁻¹), but the spatial patterns and source attributions are different. The largest
difference is for coal mining emissions in the North China Plain, where in situ observations indicates a





decrease by -0.8 Tg a$^{-1}$ a$^{-1}$ while GOSAT shows an increase by 0.1 Tg a$^{-1}$ a$^{-1}$. A previous GOSAT inversion
study found a large increase of coal mining emissions in China over 2010-2015 (Miller et al., 2019).
However, a recent bottom-up inventory estimates that Chinese coal emission peaked in 2012 and
decreased afterward, leading to no significant overall trend for 2010-2016 (Sheng et al., 2019). Our
inversion assumes linear trends in emissions over 2010-2017 but that may not be appropriate for China.

**3.5 Global methane budget for 2010-2017**
Table 1 shows the optimized global anthropogenic emissions from different sectors as determined by the
joint GOSAT + in situ inversion. Corrections to the global prior estimates are mostly determined by
GOSAT (Fig. 8). They include upward corrections to livestock and rice methane emissions, and
downward correction to the coal mining emissions driven by overestimation in China. The joint inversion
also estimates a global increase in anthropogenic emissions by 1.7±0.6 Tg a$^{-1}$ a$^{-1}$ (0.5% a$^{-1}$) in 2010-2017,
dominantly driven by trends in the tropics (Fig. 11).

A number of previous studies have analyzed surface observations to interpret global methane budgets and
trends (Dlugokencky et al., 2009; Bruhwiler et al., 2014; Houweling et al., 2017). As shown in Figure 6,
our In situ-only inversion can fit the GOSAT observations of global methane distribution and trend,
indicating that the in situ data provide useful information on the global budget. Here we examine whether
this information adds to that from GOSAT. For this purpose and following Maasakkers et al. (2019), we
collapse the full state vector to a reduced state vector ($\widehat{x}_{red}$) that contains global mean methane emissions
and OH as elements, and derive the associated error covariance matrix ($\widehat{S}_{red}$) as introduced in Section

493 2.4.


Figure 13 shows the joint probability density functions (PDFs) of the mean anthropogenic methane
emissions and methane lifetime against oxidation by tropospheric OH from the three inversions. There is
strong negative correlation ($r$=-0.72) between the optimization of methane emissions and OH in the
GOSAT-only inversion, and somewhat less in the In situ-only inversion ($r$=-0.53), although the posterior
error variance is larger due to the lower data density as indicated by the axes of the ellipses. A sensitivity
inversion using only the surface and tower measurements in the In situ-only inversion yields $r$=-0.36. It
indicates that in situ observations are more effective than the satellite observations in independently
constraining methane emissions from the sink by OH. A likely reason is that surface measurements in
source regions are more sensitive to methane emissions than are column measurements.

Comparison of the posterior PDFs between the GOSAT-only and In situ-only inversions implies that the
two are inconsistent, since the 99% probability contour does not overlap (Fig.13), but this is likely because
the posterior error covariance matrix underestimates the actual error variance in particular for global
budget errors due to its assumption of independent identically distributed (IID) observational errors
(Brasseur and Jacob, 2017). Remarkably, the GOSAT + in situ joint inversion is more in agreement with
in situ observations than GOSAT. Inspection of Figure 6c shows that the GOSAT-only inversion is biased





low relative to in situ observations at northern mid-latitudes and biased high in the southern hemisphere,
implying that both emissions and OH concentrations are too low. Ingestion of the in situ observations in
the inversion corrects that bias, and narrows the posterior error of mean anthropogenic emissions and
methane lifetime against tropospheric OH by 30% (Fig. 13), compared to the GOSAT-only inversion.
Thus we find that the GOSAT and in situ observations are complementary in quantifying the global budget.
Table 3 summarizes the global mean methane budget in 2010-2017. The GOSAT + in situ joint inversion
estimates a total methane emission of 551±2 Tg a$^{-1}$, of which 371 Tg a$^{-1}$ are anthropogenic, and a total
sink of 528±2 Tg a$^{-1}$. The total emission is at the low end of the 538-593 Tg a$^{-1}$ range reported for the
2008-2017 decade by the Global Carbon Project (Saunois et al., 2020). Our joint inversion yields a
methane lifetime against OH oxidation of 11.2±0.1 years, compared to the observationally-based estimate
of 11.2±1.3 years (Prather et al., 2012), and pushes the northern to southern hemispheric OH ratio (0.98
in GOSAT + in situ inversion versus 1.16 in prior estimate) closer to observed values (0.97±0.12) (Patra
et al., 2014).
**4 Conclusions**
We quantify and attribute global sources, sinks, and trends of atmospheric methane for 2010-2017 by
inversions of GOSAT satellite data and the GLOBALVIEWplus in situ methane observations from surface
sites, towers, ships, and aircraft. The inversions use an analytical solution to Bayesian optimization
problem including closed-form error covariance matrices from which the detailed information content of
the inversion can be derived. We conduct inversions using GOSAT and in situ data separately and
combined. In this manner we are able to quantify the consistency and complementarity (or redundancy)
of the satellite and in situ observations.
We find that the GOSAT and in situ data are generally consistent and can fit each other independently
through our inversions. Nevertheless, the GOSAT-only inversion has difficulty to fit the in situ
observations in source regions (US and Europe), while In situ-only inversions could not reproduce the
interannual variability of methane growth rate due to its heavy weighting to the northern mid-latitudes.
The GOSAT + in situ inversion shows the best agreement with observations.
GOSAT and in situ observations are to a large extent complementary in terms of constraining global
emissions. GOSAT provides stronger constraints than in situ observations for the tropics, while in situ
observations are more important in US, Canada, Europe, and northern China where observations are dense.
The joint GOSAT + in situ inversion reveals large underestimates of oil/gas emissions in the US and
Canada, and large overestimates of coal emissions in China, relative to the national inventories reported
to the United Nations Framework Convention on Climate Change (UNFCCC) and used here as prior
estimates for our inversions. Emissions from boreal wetlands are overestimated in the mean WetCHARTs
inventory used as prior estimate, particularly in May-June when snow cover and frozen soils inhibit
methane emission.



Our inversions estimate increasing trends in US anthropogenic emissions driven by oil/gas production but decreasing trends in Canada (oil/gas) and Europe. Joint inversion of GOSAT and in situ data show weak decreasing trend in Chinese coal emissions for 2010-2017, consistent with a recent bottom-up inventory (Sheng et al., 2019).

We find that GOSAT and in situ observations are also complementary in constraining global methane budget. While the global budget information relies more on GOSAT observations, information from the in situ observations at northern mid-latitudes avoids the large error correlations between methane emissions and sink from OH and also corrects the underestimation of both emission and OH in the GOSAT-only inversion. Our joint GOSAT + in situ inversion yields the global methane emissions and loss of $551\pm2$ and $528\pm2$ Tg $a^{-1}$ $a^{-1}$ averaged over 2010-2017, and methane lifetime of $11.2\pm0.1$ years.

Our study presents a framework to integrate satellite and in situ data in analytical inversions. We conclude that on the basis of the present observation system, in situ and satellite observations are complementary for constraining global methane budgets and regional emissions. Satellite observations of atmospheric methane are presently expanding with the new availability of global daily data from the TROPOMI instrument launched in October 2018 (Hu et al., 2018). In situ observations as presented in this paper will continue to play a critical role for satellite validation and for quantification of long-term trends. Their role for source characterization in supplement to satellite data will need to be re-evaluated as satellite observations expand, and the framework presented in this paper provides a means for doing so.

**Data availability**

The GLOBALVIEWplus CH$_4$ ObsPack v1.0 data product is available at https://www.esrl.noaa.gov/gmd/ccgg/obspack/data.php?id=obspack_ch4_1_GLOBALVIEWplus_v1.0_2019-01-08 (last access: July 17, 2020). The GOSAT proxy satellite methane observations are available at https://doi.org/10.5285/18ef8247f52a4cb6a14013f8235cc1eb (last access: July 17, 2020). Modeling data can be accessed by contacting the corresponding authors Xiao Lu (xiaolu@g.harvard.edu) and Yuzhong Zhang (zhangyuzhong@westlake.edu.cn).

**Author contributions**

XL and DJJ designed the study. XL and YZZ conducted the modeling and data analyses with contributions from JDM, MPS, LS, ZQ, TRS, HON, RMY, and JXS. AA contributed to the GLOBALVIEWplus CH$_4$ ObsPack v1.0 data product. RJP and HB contributed to the GOSAT satellite methane retrievals. AAB and SM contributed to the WetCHARTs wetland emission inventory and its interpretation. XL and DJJ wrote the paper with input from all authors.

**Competing interests**

The authors declare that they have no conflict of interest.



**Acknowledgement**

This work was supported by the NOAA AC4 program. RJP and HB are funded via the UK National Centre for Earth Observation (NCEO grant numbers: NE/R016518/1 and NE/N018079/1). RJP and HB acknowledge funding from the ESA GHG-CCI and Copernicus C3S projects. We thank the Japanese Aerospace Exploration Agency, National Institute for Environmental Studies, and the Ministry of Environment for the GOSAT data and their continuous support as part of the Joint Research Agreement. This research used the ALICE High Performance Computing Facility at the University of Leicester for the GOSAT retrievals. Part of this research was carried out at the Jet Propulsion Laboratory, California Institute of Technology, under a contract with the National Aeronautics and Space Administration.

We acknowledge all data providers/laboratories (https://search.datacite.org/works/10.25925/20190108) contributed to the GLOBALVIEWplus $CH_4$ ObsPack v1.0 data product compiled by NOAA Global Monitoring Laboratory. We acknowledge methane observations collected from the CONTRAIL (Comprehensive Observation Network for TRace gases by AIrLiner) project (Machida et al., 2019). Data collected at WLEF Park Falls towers were supported by the NSF DEB-0845166 and DOE Ameriflux Network Management Project. Data collected at the Southern Great Plains were supported by the Office of Biological and Environmental Research of the US Department of Energy under contract no. DE-AC02-05CH11231 as part of the Atmospheric Radiation Measurement (ARM) Program, ARM Aerial Facility (AAF), and Terrestrial Ecosystem Science (TES) Program.

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





**Table 1.** Global sources and sinks of atmospheric methane, 2010-2017[a].

|  | Prior [b] | Posterior [c] |
|---|---|---|
| **Total sources [Tg a$^{-1}$]** | 533 | 551 |
| **Natural Sources** |  |  |
| Wetlands | 161 | 148 |
| Open fires | 14 | 16 |
| Termites | 12 | 14 |
| Seeps | 2 | 2 |
| **Anthropogenic sources** |  |  |
| Livestock | 117 | 136 |
| Oil | 42 | 40 |
| Natural gas | 25 | 30 |
| Coal mining | 31 | 23 |
| Rice cultivation | 38 | 44 |
| Wastewater | 37 | 42 |
| Landfills | 30 | 31 |
| Other Anthropogenic | 25 | 25 |
| **Total Sinks [Tg a$^{-1}$]** | 540 | 528 |
| Tropospheric OH | 468 | 455 |
| Stratospheric loss [d] | 33 | 33 |
| Soil uptake [d] | 34 | 34 |
| Tropospheric Cl [d] | 5 | 5 |

[a] 8-year mean values for 2010-2017.
[b] Prior natural source estimates (2000-2017 means) are from Bloom et al. (2017) for wetlands, Etiope et al. (2019) and
Hmiel et al. (2020) for seeps, Fung et al. (1991) for termite emissions, van der Werf et al. (2017) for open fire emissions.
Prior anthropogenic source estimates for 2012 are from EDGAR v4.3.2 (Janssens-Maenhout et al., 2017) except from
Scarpelli et al. (2020) for fuel exploitation (oil, gas, coal), and are overwritten for the US with the gridded EPA inventory
of Maasakkers et al. (2016). The prior tropospheric OH concentration field is from Wecht et al. (2014) and yields a
methane lifetime against oxidation by tropospheric OH of 10.6 years.
[c] From the joint inversion of GOSAT and in situ data
[d] These minor sinks are not optimized by the inversion.





**Table 2.** Anthropogenic methane emissions and trends, 2010-2017 [a]

| Inversions | In situ-only inversion | GOSAT-only inversion | GOSAT+in situ inversion |
|---|---|---|---|
| US [b] (prior: 28 Tg a$^{-1}$) | | | |
| Posterior (Tg a$^{-1}$) | 35 | 31 | 36 |
| 2010-2017 trend (Tg a$^{-1}$ a$^{-1}$) | 0.5 | -0.1 | 0.4 |
| | | | |
| Canada (prior: 5 Tg a$^{-1}$) | | | |
| Posterior (Tg a$^{-1}$) | 8 | 5 | 8 |
| 2010-2017 trend (Tg a$^{-1}$ a$^{-1}$) | -0.2 | -0.0 | -0.2 |
| | | | |
| Europe [c] (prior: 27 Tg a$^{-1}$) | | | |
| Posterior (Tg a$^{-1}$) | 28 | 17 | 23 |
| 2010-2017 trend (Tg a$^{-1}$ a$^{-1}$) | 0.1 | -0.6 | -0.4 |
| | | | |
| China (prior: 63 Tg a$^{-1}$) | | | |
| Posterior (Tg a$^{-1}$) | 45 | 46 | 43 |
| 2010-2017 trend (Tg a$^{-1}$ a$^{-1}$) | 0.3 | 0.4 | 0.1 |

[a] Posterior estimates of mean 2010-2017 emissions and trends for the In situ-only, GOSAT-only, and GOSAT + in situ
joint inversions.
[b] Including contiguous US and Alaska.
[c] Europe is defined as west of 30ºE, excluding Russia.





**Table 3.** Optimized global methane budget, 2010-2017.

| Inversions | In situ | GOSAT | GOSAT+in situ |
|---|---|---|---|
| **Total sources [Tg a⁻¹]** | $515\pm4^d$ | $504\pm3^d$ | $551\pm2^d$ |
| Anthropogenic [a] | 359 | 333 | 371 |
| Seeps, termites | 15 | 15 | 16 |
| Open fires | 15 | 16 | 16 |
| Wetlands | 126 | 140 | 148 |
| **Total sinks [Tg a⁻¹]** | $494\pm4^e$ | $478\pm3^e$ | $528\pm2^e$ |
| Tropospheric OH[b] | 421 | 406 | 455 |
| Other losses [c] | 73 | 72 | 73 |
| **Mean imbalance [Tg a⁻¹]** | 21 | 26 | 23 |

[a] See Table 1 for sectoral breakdown from the joint inversion.
[b] Methane lifetime against oxidation by tropospheric OH is 11.2±0.1 years in the GOSAT + in situ inversion.
[c] Soils, stratosphere, and oxidation by tropospheric Cl.
[d] Error standard deviation estimated from the quadrature of error variance of non-wetland emissions and wetland
emissions.
[e] Error standard deviation only accounts for the uncertainty in oxidation by tropospheric OH.





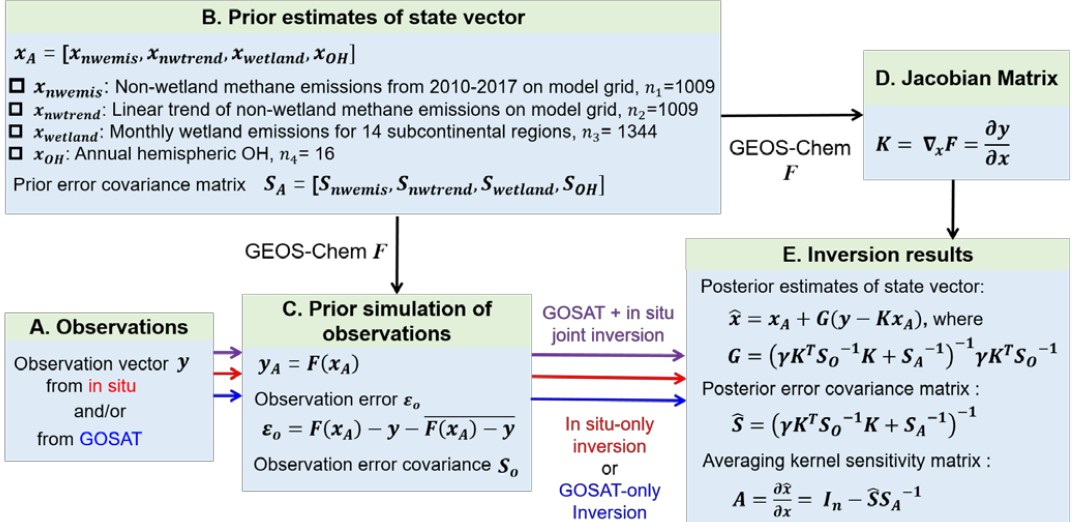

**Figure 1.** Analytical inversion framework. The inversion is applied to GOSAT and GLOBALVIEWplus
in situ observations for 2010-2017. GEOS-Chem is the chemical transport model (CTM) used as forward
model for the inversion. $\gamma$ is a regularization factor in the Bayesian cost function (see text).



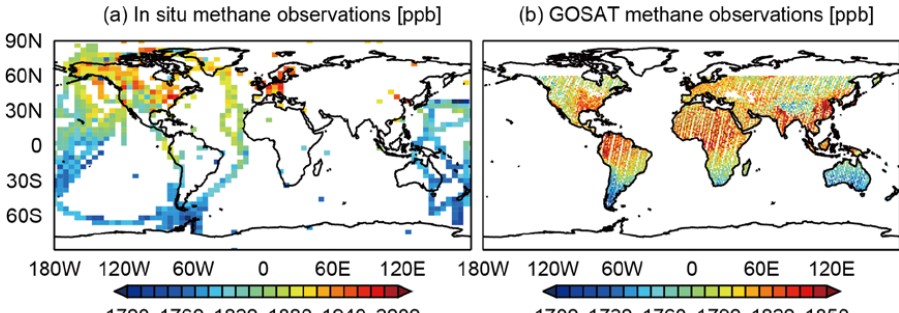

**Figure.2** Mean 2010-2017 methane observations from GLOBALVIEWplus ObsPack data product and
GOSAT. The GLOBALVIEWplus in situ data are local dry mixing ratios and are averaged over the 4°×5°
model grid for visibility. The GOSAT data are dry column mixing ratios on a 1°×1° grid from the
University of Leicester version 9 Proxy XCH$_4$ retrieval (Parker et al., 2020), excluding observations over
oceans and poleward of 60°N. Note the difference in color scale between panels.





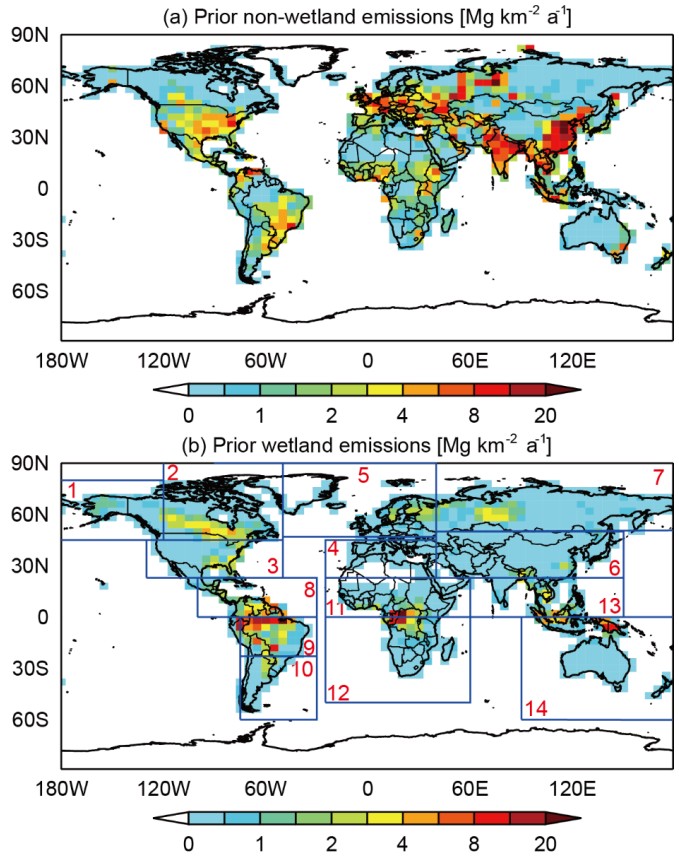

**Figure 3.** Prior estimates of mean 2010-2017 methane emissions. The top panel shows the non-wetland emissions on the 4°×5° grid used for the inversion. The bottom panel shows the wetland emissions and the 14 subcontinental wetland regions used for the inversion following Bloom et al. (2017).

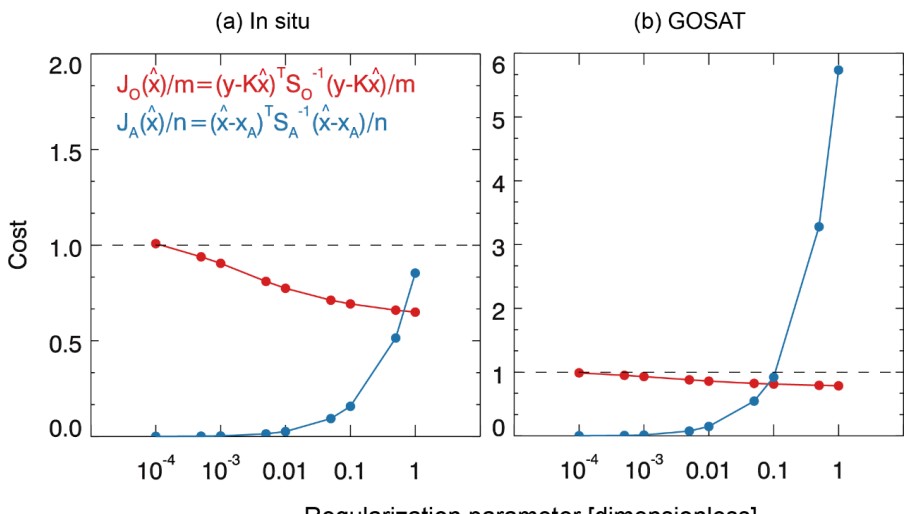

**Figure 4.** Optimization of the regularization parameter $\gamma$ in the Bayesian cost function (Equation (1)). The figure shows the posterior observation component $J_O(\hat{x}) = (y - K\hat{x})^T S_O^{-1}(y - K\hat{x})$ and the posterior state component $J_A(\hat{x}) = (\hat{x} - x_A)^T S_A^{-1}(\hat{x} - x_A)$ for the In situ-only and GOSAT-only.

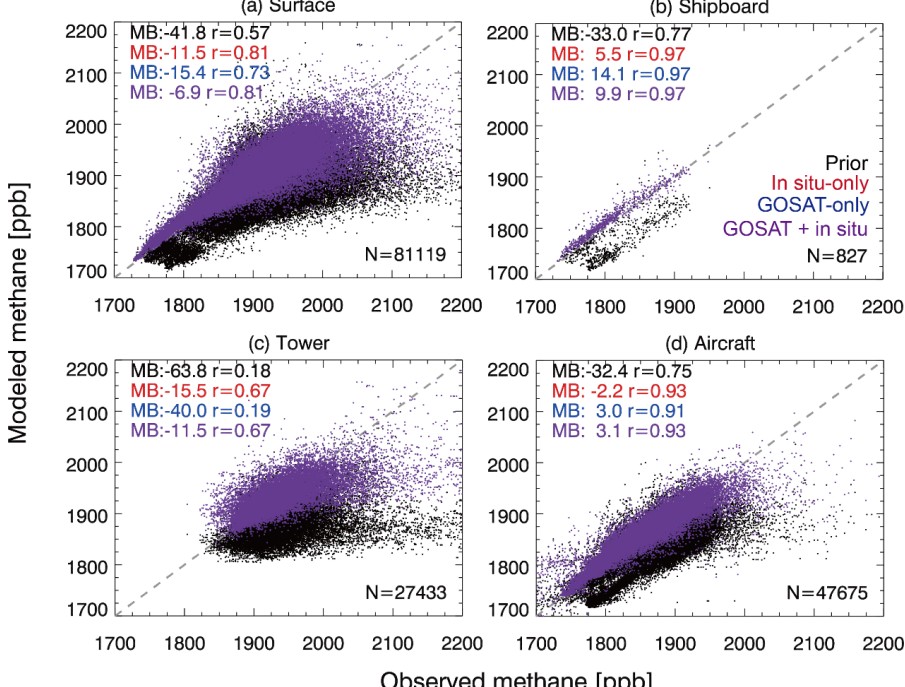

**Figure 5.** Ability of the inversions to fit the in situ methane observations. Panels (a)-(d) compare the surface, tower, shipboard, and aircraft observations in 2010-2017 to the GEOS-Chem simulation using the prior (black) and posterior estimates of methane emissions and OH concentrations from the In situ-only inversion (red, dots not shown), GOSAT-only inversion (blue dots not shown), and GOSAT + in situ joint inversion (purple). The numbers (N) of observations from each platform, the mean bias (MB), and the correlation coefficients (*r*) between the observed and simulated values are shown inset.





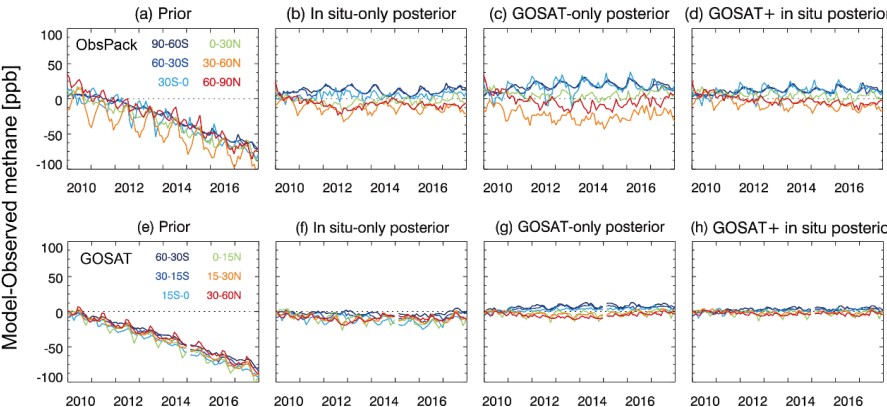

954
955

**Figure 6.** Ability of the inversions to fit the in situ methane observations and GOSAT satellite
observations. Panels (a)-(d) show the monthly time series of the differences between observed and
simulated in situ methane concentrations averaged over different latitude bands from 2010 to 2017. Panels
(e)-(h) are the same as panels (a)-(d) but for GOSAT methane concentrations.

960
961





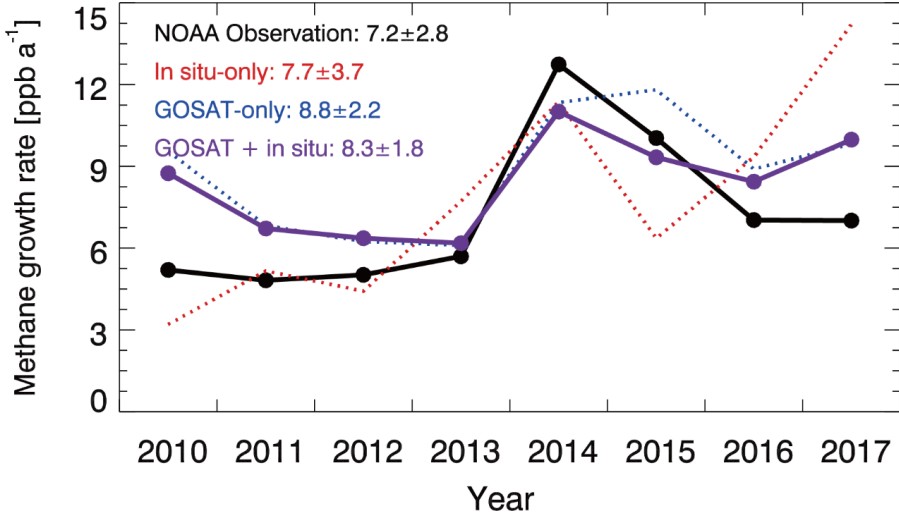

**Figure 7.** Annual global growth rate of atmospheric methane, 2010-2017. Results from our three different inversions (In situ-only, GOSAT-only, GOSAT + in situ) are compared to the observed growth rates inferred from the NOAA surface observational network (https://www.esrl.noaa.gov/gmd/ccgg/trends_ch4/, last access: 20 June, 2020). Mean annual growth rates and standard deviations from the different inversions are shown inset.






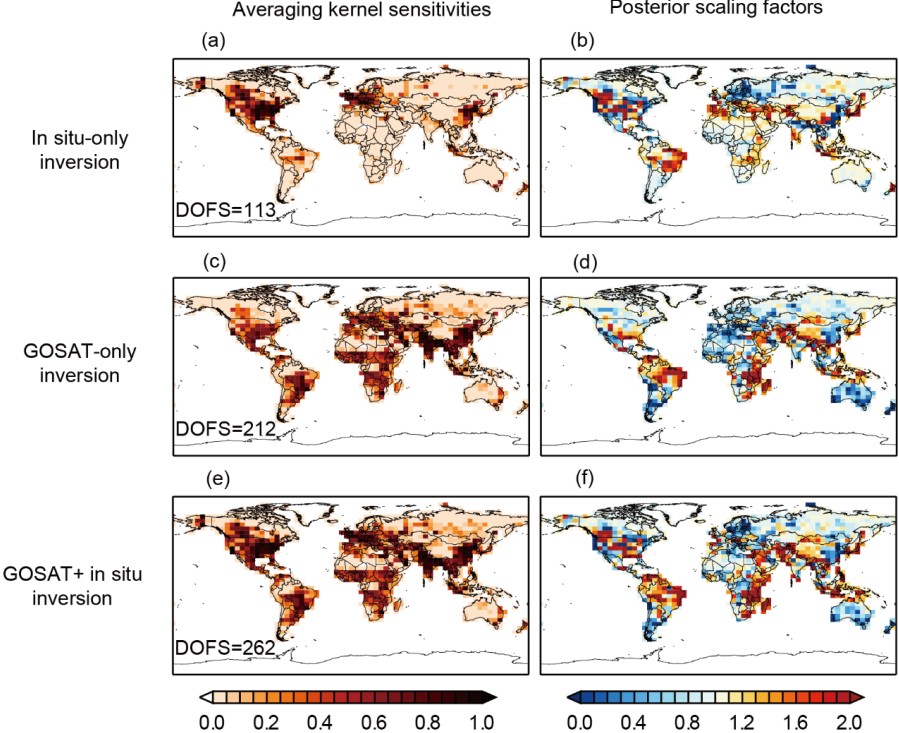



**Figure 8.** Optimization of mean 2010-2017 non-wetland (mainly anthropogenic) emissions. The In situ-
only inversion uses in situ observations, the GOSAT-only inversion uses GOSAT satellite observations,
and the GOSAT + in situ inversion uses both. The left panels show the averaging kernel sensitivities
(diagonal elements of the averaging kernel matrix) for each inversion, with the degrees of freedom for
signal (DOFS, defined as the trace of the averaging kernel matrix) given inset. The right panels show the
correction factors to the prior emissions (Figure 3a). Wetland emissions are corrected separately (see text).

**Figure 9.** Optimization of anthropogenic methane emissions by source sectors in the In situ-only, GOSAT-only, and GOSAT + in situ inversions. The left panel shows the averaging kernel sensitivities for each emission sector (see text for description), the right panel shows the emissions. Europe is defined as west of 30°E, which excludes Russia.



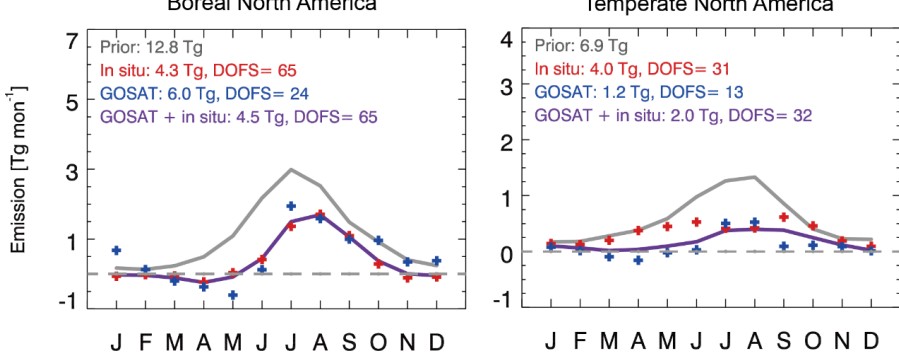

**Figure 10.** Wetland emissions in boreal and temperate North America (regions 2 and 3 of Figure 3). Prior
and posterior estimates of the monthly mean wetland emissions averaged over 2010-2017 from different
inversions are shown. Annual mean emissions and the degree of freedom for signal (DOFS) are shown
inset. Note differences in scale between panels. Negative emissions are allowed statistically by the
inversion but are likely not physical.

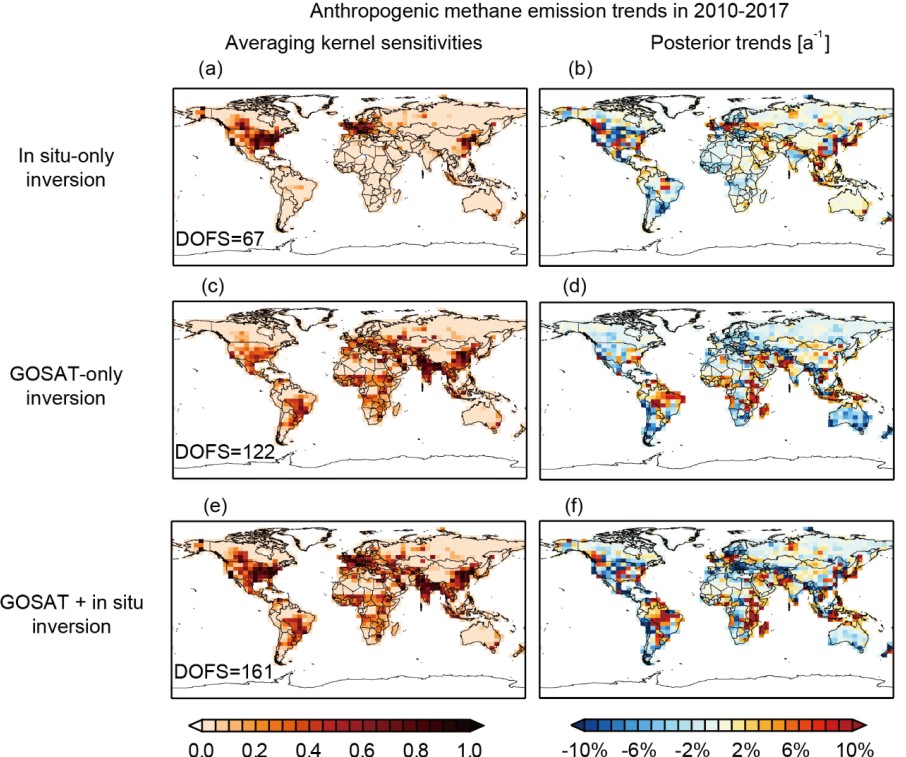

**Figure 11.** Same as Figure 8 but for optimization of non-wetland (mainly anthropogenic) emission trends in 2010-2017.



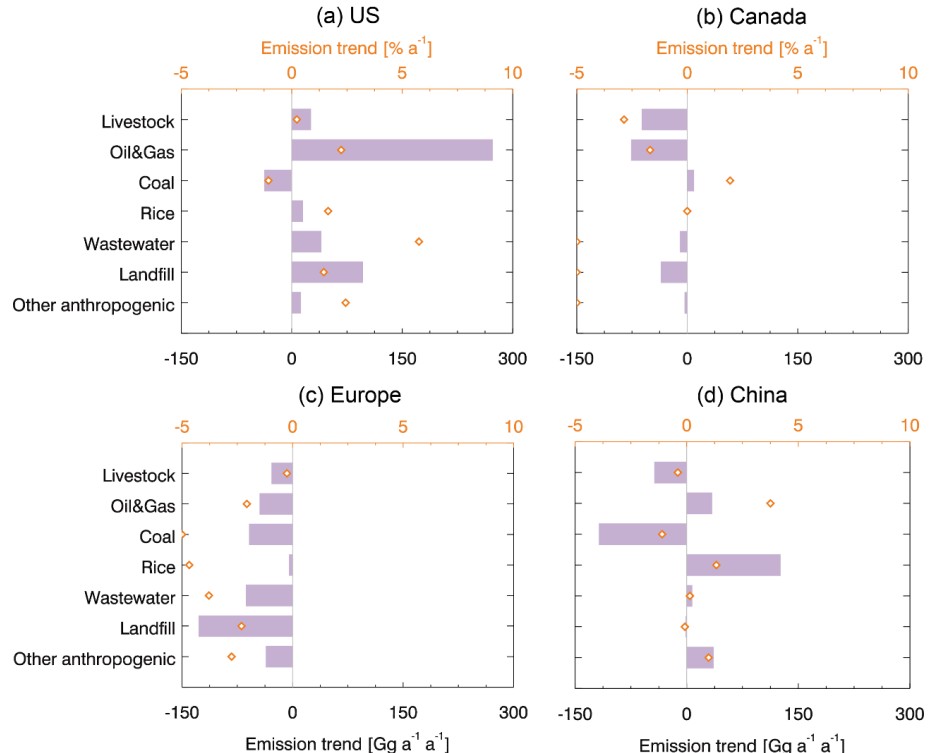


**Figure 12.** Optimization by sector of regional anthropogenic methane emission trends in 2010-2017. Bars
and diamonds represent trends in Gg a$^{-1}$ a$^{-1}$ (bottom axis) and % a$^{-1}$ (top axis) over the 2010-2017 period
from the GOSAT + in situ joint inversion.







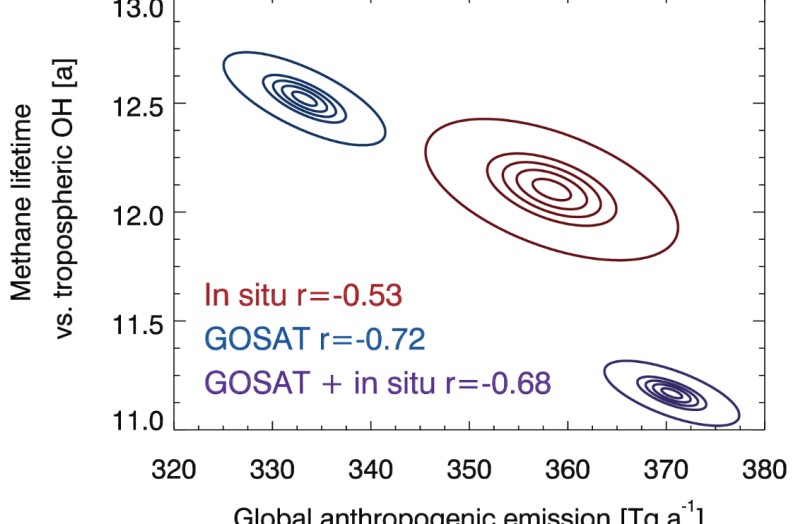


**Figure 13.** Joint probability density functions (PDFs) of global mean anthropogenic methane emission and methane lifetime against oxidation by tropospheric OH optimized by the three inversions. The thick contours show probabilities of 0.99 (outermost), 0.7, 0.5, 0.3, and 0.1 (innermost) from the three base inversions. The error correlation coefficients are given inset.