# Peer review of "Global methane budget and trend, 2010-2017: complementarity of inverse analyses using in situ (GLOBALVIEWplus CH$_4$ ObsPack) and satellite (GOSAT) observations"

_Atmospheric Chemistry and Physics, 2020_

## Short Comment (SC1) · 3 Nov 2020

**Short comment to Lu et al.:**
**Global methane budget and trend, 2010–2017: complementarity of inverse analyses using in situ (GLOBALVIEWplus CH$_4$ ObsPack) and satellite (GOSAT) observations**

Luke Western[1]

[1]Atmospheric Chemistry Research Group, University of Bristol, Bristol, UK

October 2020

**1   Reason for comment**

I write this short comment to discuss a minor part of the discussion paper by Lu et al. (2020), in which a statistical condition is erroneously interpreted. This condition erroneously appears in many other sources of literature, and the belief in this condition has seemed to be confounded as a result.

The condition is that in equations (6) and (7) of Lu et al. (2020), which states that

$$J_A(\hat{\mathbf{x}}) = (\hat{\mathbf{x}} - \mathbf{x}_A)^T \mathbf{S}_A^{-1} (\hat{\mathbf{x}} - \mathbf{x}_A) \approx n, \tag{1}$$

and

$$J_O(\hat{\mathbf{x}}) = (\mathbf{y} - \mathbf{K}\hat{\mathbf{x}})^T \mathbf{S}_O^{-1} (\mathbf{y} - \mathbf{K}\hat{\mathbf{x}}) \approx m, \tag{2}$$

using the variables in Lu et al. (2020). These conditions state that the sum of log-likelihood and log-prior terms in the 'cost function' should be approximately equal to the number of observations, $m$ for $\mathbf{y}$, or inferred parameters, $n$ for $\mathbf{x}$, respectively at the maximum a posteriori value of $\mathbf{x}$.

The paper elaborates on this condition, for example for the component concerning the prior distribution, saying that "$J_A(\hat{\mathbf{x}}) >> n$ implies overfit to the observations because the posterior state vector estimates are far outside the estimated errors on the prior estimates." In addition, there is the statement "In our case the prior error covariance matrix is not strictly diagonal because of covariance for the wetland terms (Bloom et al., 2017), so one may expect $J_A(\hat{\mathbf{x}})$ to be somewhat deviated from $n$."

I hope in the following that I will demonstrate that these statements have no foundations in Bayesian probability theory, and likely have become pervasive due to an earlier misinterpretation of the mathematics, and subsequent adoption of this. I will explain what the mathematics show with respect to the error distribution of such distributions.

**2   Properties of the Multivariate Normal**

My assumption is that the confusion has stemmed from a misinterpretation of the condition outlined in, for example, Tarantola (2005), Ch6, which discusses the application of the Chi-squared distribution. It is worth noting that there are many other texts with a more mathematical description of this concept (e.g. Mardia et al., 1979). We can apply the properties of the Chi-squared distribution in Tarantola (2005), using less ambiguous notation, to the problem as framed in Lu et al. (2020), for example to the likelihood, where

$$J_O(\mathbf{x}) = (\mathbf{y} - \mathbf{K}\mathbf{x})^T \mathbf{S}_O^{-1} (\mathbf{y} - \mathbf{K}\mathbf{x}), \tag{3}$$

where the random variable $J_O(\mathbf{x})$ is distributed for all possible values according to the $\chi^2$ distribution with

$$\nu = \dim(\mathbf{y}) = m \tag{4}$$

degrees of freedom. Note here that it doesn't say that $J_O(\mathbf{x}) = \nu$, nor $J_O(\mathbf{x}) \approx \nu$, but $J_O(\mathbf{x}) \sim \chi^2_\nu$, i.e. it is distributed with this distribution.

This is where I assume much of the confusion has come from. The earliest erroneous statement that I can find is in Michalak et al. (2005), but there may be others before this. Note also that the presence or lack of off-diagonal elements in the covariance matrix makes no difference to the statement in equation 3 and its subsequent distribution.

So what does $J_O(\mathbf{x}) \sim \chi^2_\nu$ mean practically? In a frequentist (i.e. non-Bayesian) setting, the 'cost' corresponds to a particular probability contour, following the quantile function of the Chi-squared distribution. Figure 1 shows a toy frequentist 'inversion' of two parameters, shown by the values on the x and y axis. The true values are 1 and 2. These were informed using 5 observations. The coloured background shows the 'cost' over the parameter space and the contours show the corresponding probability content according to the Chi-squared distribution. This has a practical application, for example to define the uncertainty in an estimated value (see e.g. Western et al., 2020).

[Figure]

Figure 1: A toy frequentist 'inversion' of two parameters, shown by the values on the x and y axis. The true values are 1 and 2. These was informed using 5 observations. The coloured background show the 'cost' at the parameter values and the contours show the corresponding probability content according to the Chi-squared distribution.

This idea can also be readily applied, for example to equation 2. In a frequentist setting, if $J_O(\hat{\mathbf{x}}) = m$, this also has a corresponding probability following the quantile function of the Chi-squared distribution. That is, all values of $\mathbf{x}$ which are less or equal to some value of $J_O(\mathbf{x})$ can be translated into a confidence region with a defined probability. For example, if $m = 100$, for all values of $\mathbf{x}$ where $J_O(\mathbf{x}) \leq 124.3421$, then all these values fall within the 95% confidence region of the maximum likelihood estimate, or we can say with 95% certainty that the 'true' value falls within this parameter space. Figure 2 shows this probability for $J_O(\hat{\mathbf{x}}) = m$ for $1 \leq m \leq 100$. Or, in other words, Figure 2 shows the probability contour of a confidence region at $J_O(\hat{\mathbf{x}}) = m$, for a problem with $m$ degrees of freedom. This probability asymptotes for large $m$, but what is key is that this probability content at $m$ (or $n$) changes depending on the degrees of freedom.

Therefore, unless $m$ and $n$ are equal, or at least both very large, equations 1 and 2 are not making a comparison to the same probability. If $m = 1000$, then this probability contour for all values of $J_O(\hat{\mathbf{x}}) \leq m$ is around 51%, whereas if $n = 10$, all values where $J_A(\hat{\mathbf{x}}) \leq m$ is around 56%. I do not see a reason why (even assuming $m = n$) it is supposed that each term should be evaluated with a probability content $\sim 0.5$ at $\hat{\mathbf{x}}$. If $J_A(\hat{\mathbf{x}}) < n$, and $J_O(\hat{\mathbf{x}}) < m$, why would this suggest an overfit?

[Figure]

Figure 2: The probability contour at $J_O(\hat{\mathbf{x}}) = m$ according to the quantile function of the Chi-squared distribution with $m$ degrees of freedom.

**3    Does any of this matter?**

The reason I am talking about uncertainty regions is that this seems to be implicit in the concept applied. My interpretation of, for example, equation 2 (Lu et al., 2020, equation 7), is that if $J_O(\hat{\mathbf{x}}) \leq m$ in equation 2, one would assume that the inversion is over confident in its estimated value, and hence the uncertainty is smaller than it should be. In Figure 1, this would translate as the contours on the plot being much smaller than they should be – the results are showing too much confidence in the inversion's estimates. This makes intuitive sense (even if equations 1 and 2 do not make sense statistically). However, a problem with the discussion in Section 2 is that the connection to uncertainty regions is valid for frequentist statistics, but not for Bayesian statistics, which is the stated approach to inference taken. Instead, measures of uncertainty in Bayesian inference rely on integration over the parameter space, which results in a fixed interval in which the 'truth' resides, as opposed to the uncertainty about a fixed most probable value in frequentist statistics. This means that the derived uncertainty is not the uncertainty in $\hat{\mathbf{x}}$ (a frequentist idea), but rather a fixed uncertainty region for $\mathbf{x}$ in which some metric $\hat{\mathbf{x}}$ resides. An example of a suitable Bayesian uncertainty region is the Highest Posterior Density (see e.g. Box and Tiao, 1992, Ch.2), defined as the narrowest region, $R$, in the total posterior parameter space that holds probability content $(1 - \alpha)$, or

1. $\mathrm{p}\{\mathbf{x} \in R\} = (1 - \alpha)$
2. for $\mathbf{x}_1 \in R$ and $\mathbf{x}_2 \notin R$, $\mathrm{p}(\mathbf{x}_1 \mid \mathbf{y}) \leq \mathrm{p}(\mathbf{x}_2 \mid \mathbf{y})$.

Although, in the case presented in the paper due to the Gaussian likelihood and prior, and resultant Gaussian posterior, such an integration is simple and readily available.

In my opinion, an improvement on trying to *post hoc* adjust probability distributions can take one of two paths. The first is to explicitly include uncertainty in parameters within the inversion itself, following either an empirical Bayes or hierarchical approach (e.g. Michalak et al., 2005; Ganesan et al., 2014), and thus formally considering the probabilities. The second is to invest some time in creating a better prior probability distribution that is representative of your actual prior belief of the possible parameter space. See e.g. Rougier (2007), Sect 2, for a more thorough discussion on this topic.

The second suggestion raises an interesting question – is the inversion approach taken in this work, and many others, actually probabilistic or is it a regularisation but explained using concepts from Bayesian probability? This has previously been raised in the context of remote sensing by e.g. Cressie (2018). The adjustment of 'probability distributions' to better fit models using the concept of a 'cost function' in my opinion falls closer to a regularisation problem. That is, if your posterior probability indicates that the mean inferred parameters have a low probability according to your prior probability, then this does not mean that the posterior/prior is wrong, and you may miss low-frequency events by removing this. If this happens consistently, then of course some reevaluation of the model or prior knowledge should take place. Using *post hoc* adjustment instead gives the impression that the prior probability (its functional form and parameter values), the uncertainties in the likelihood and the use of the extra variable $\gamma$, are instead weightings given to guide an optimisation procedure. The use of a the regularisation factor $\gamma$ (as used in Lu et al. (2020)) in inverse modelling comes from regularisation rather than anything probabilistic (Tikhonov, 1963), which is somewhat 'un-Bayesian' in its current application unless included within the probabilistic hierarchy. Regularisation is fine – the machine learning community in particular has had a lot of success in working with optimisation through regularisation – but it then means that concepts such as uncertainty in the posterior estimate is not probabilistic and as such is difficult to interpret. As a result, the approach taken in the work should probably not be described as Bayesian, or probabilistic.

**4   Final remarks**

I do not want this to seem like an attack on the paper – it is not. In fact, I think the paper is very good and hence a suitable platform to raise this issue (rather than some work which generally has more pressing issues). I commend the work that has been done and hope for its eventual publication.

I have also been purposefully slightly provocative in my arguments, in order to facilitate discussion, which I hope others in the community will contribute to – including the nominated reviewers. I am willing to be proved wrong in my arguments, and indeed welcome a proof of the statement that has thus far, to my knowledge, not been sufficiently presented, even as a heuristic.

**References**

Bloom, A. A., Bowman, K. W., Lee, M., Turner, A. J., Schroeder, R., Worden, J. R., Weidner, R., McDonald, K. C., and Jacob, D. J. (2017). A global wetland methane emissions and uncertainty dataset for atmospheric chemical transport models (WetCHARTs version 1.0). *Geoscientific Model Development*, 10(6):2141–2156.

Box, G. E. P. and Tiao, G. C. (1992). *Bayesian inference in statistical analysis*. A Wiley-Interscience publication. Wiley, New York, wiley classics library ed edition. OCLC: 25247039.

Cressie, N. (2018). Mission CO $_2$ ntrol: A Statistical Scientist's Role in Remote Sensing of Atmospheric Carbon Dioxide. *Journal of the American Statistical Association*, 113(521):152–168.

Ganesan, A. L., Rigby, M., Zammit-Mangion, A., Manning, A. J., Prinn, R. G., Fraser, P. J., Harth, C. M., Kim, K.-R., Krummel, P. B., Li, S., Mühle, J., O'Doherty, S. J., Park, S., Salameh, P. K., Steele, L. P., and Weiss, R. F. (2014). Characterization of uncertainties in atmospheric trace gas inversions using hierarchical Bayesian methods. *Atmospheric Chemistry and Physics*, 14(8):3855–3864.

Lu, X., Jacob, D. J., Zhang, Y., Maasakkers, J. D., Sulprizio, M. P., Shen, L., Qu, Z., Scarpelli, T. R., Nesser, H., Yantosca, R. M., Sheng, J., Andrews, A., Parker, R. J., Boech, H., Bloom, A. A., and Ma, S. (2020). Global methane budget and trend, 2010–2017: complementarity of inverse analyses using in situ (GLOBALVIEWplus CH4 ObsPack) and satellite (GOSAT) observations. preprint, Gases/Atmospheric Modelling/Troposphere/Physics (physical properties and processes).

Mardia, K. V., Kent, J. T., and Bibby, J. M. (1979). *Multivariate analysis*. Probability and mathematical statistics. Academic Press, London ; New York.

Michalak, A. M., Hirsch, A., Bruhwiler, L., Gurney, K. R., Peters, W., and Tans, P. P. (2005). Maximum likelihood estimation of covariance parameters for Bayesian atmospheric trace gas surface flux inversions. *Journal of Geophysical Research: Atmospheres*, 110(D24).

Rougier, J. (2007). Probabilistic Inference for Future Climate Using an Ensemble of Climate Model Evaluations. *Climatic Change*, 81(3-4):247–264.

Tarantola, A. (2005). *Inverse problem theory and methods for model parameter estimation*. Society for Industrial and Applied Mathematics, Philadelphia, PA. OCLC: ocm56672375.

Tikhonov, A. N. (1963). Solution of incorrectly formulated problems and the regularization method. *Soviet Math. Dokl.*, 4:1035–1038.

Western, L. M., Rougier, J. C., Watson, I. M., and Francis, P. N. (2020). Evaluating nonlinear maximum likelihood optimal estimation uncertainty in cloud and aerosol remote sensing. *Atmospheric Science Letters*, 21(8).

---

## Referee Comment (RC1) · Julia Marshall (Referee) · 5 Nov 2020

This paper presents an analysis of the global methane budget and trend from 2010-2017 by simultaneously optimizing the source distributions, the OH sink (through hemispheric scaling factors), and linear trends using an analytical inversion approach with the GEOS-Chem model. Overall it is clearly written and structured and the figures are sufficiently clear and complete. From the subject matter it fits well within the scope of ACP.

[Figure]

At first glance this paper seems extremely similar in approach and content to Maasakkers et al. (2019) who used a very similar setup with the same model over an overlapping period (2010-2015) to do basically the same thing. The main difference that I can see is that here surface measurements are also included as a data constraint in order to show their complementarity (and consistency).

There's something a bit worrying showing up in Figure 6. Figure 6 seems to show that the both the in-situ-only and GOSAT-only inversions overestimate concentrations in the southern hemisphere and underestimate them in the northern hemisphere (more in the mid-latitudes in NH than in the Arctic). Interestingly, this consistent latitude-dependent bias does not seem to be present in the priors, or at least not as strongly. (Note that the 60-90N and 60-90S curves are more or less on top of each other when compared to the observations for the prior runs.) The fact that they then diverge so systematically after optimisation seems to imply that something is going wrong with the OH hemispheric optimisation - or is there another explanation?

Interestingly this pattern appears least distinct when considering the in-situ-only posterior sampled at GOSAT locations, whereas it is most pronounced in the GOSAT-only posterior. Can you explain this? Does this have something to do with the seasonal latitudinal coverage of the GOSAT measurements? In the comparison of the GOSAT-informed concentrations (both with and without the in-situ data) to the ObsPack measurements (panels 6c and 6d, less evident in 6b) there seems to be almost an temporal anti-correlation in the model-data mismatch between the 30-60N stations and the 60-90N stations.

It seems to represent a systematic error in the interhemispheric gradient, which can be explained through either the distribution of the sink, the distribution of the sources, or errors in the transport – or most likely a combination of all three. However as both the sink and the sources are being optimised, it seems surprising that such a zonally-dependent offset is emerging. Even if there are transport errors (and there always are), I would expect a solution to emerge that was consistent with the interhemispheric

gradient of the measurements. Of course the OH sink is only being optimised as a hemispheric scaling: might this reflect a problem in the spatial or temporal distribution that is being scaled? Still, usually the fluxes will adapt to compensate, provided they have sufficient flexibility. The fact that Zhang et al. (2018) found the inversion results to be not so sensitive to different OH fields suggests that this is not the case.

Some explanation of the source of this systematic error should be included. The only mention of transport errors is the claim that the regularisation factor gamma should help account for error correlations in the observations due to transport and source aggregation errors. Interestingly this does not seem to appear in the very similar simulations from the same group with a similar set-up, as seen in Figure 3d of Maasakkers et al. (2019).

Perhaps the most interesting (while also troubling) result is in Figure 13: the negative correlation between methane lifetime and estimated (anthropogenic) emissions is not in and of itself surprising. What is surprising is the fact that none of the three solutions are in any way consistent with each another. This can be explained by an underestimating of the posterior error covariances, as the authors do in L505-509. The fact that the GOSAT+in situ result does not lie somehow between the GOSAT-only and in-situ-only result is, however, worrying. The authors suggest that this is due to a correction of a bias in the GOSAT-only inversion by ingesting the in-situ measurements. This bias was diagnosed as being in both the OH (too low, because the methane in the SH was overestimated) and the fluxes (too low, because the methane in the NH mid-latitudes was underestimated). From this perspective it makes some sense that it would correct in the direction that it did, but why would it overshoot the in-situ-only solution? Is there some fundamental inconsistency in the two types of measurements (or an error in the model) that makes it impossible to match them both simultaneously?

This result seems to suggest that the measurements themselves are not really consistent with each other, which the paper claimed to set out to test (L91-94). Thus this result seems to contradict the conclusion that "the GOSAT and in situ data are generally consistent and can fit each other independently through our inversions" (L535-536). Even if the concentrations in the different inversion come closer to each other, is the result really consistent if the emissions and the lifetime are so very divergent?

While trying to understand this rather surprising result I realised that I would like to see some more figures: OH was scaled per hemisphere per year (16 state vector values). A time series of these scaling factors (perhaps as an additional panel or two in Figure 7?) would be interesting to see, rather than just an average lifetime over the whole period (similar to Figure 7d in Maasakkers et al. (2019)). This might also help convince me that scaling OH based on surface-based methane measurements alone makes sense - do the OH scaling factors in this case stay close to one throughout?

Another plot that might help convince the reader of the adequacy of the transport model and the improvement of the sources and sinks would be geographical (zonal + altitude?) plot of the model-data mismatch for aircraft data presented in Figure 5d. Even if it has to go into a supplement, it would be a useful piece of information for the reader to assess if this very surprising result might make sense.

Once these concerns are addressed, I think the paper would be appropriate for publication in ACP.

Postscript: I have only just now seen the short comment by Western, and do not have time to read deeply into this at the moment, but wanted to submit my review all the same in the meantime.

Minor comments:

I would recommend adding how many independent pieces of information are contained in the GLOBALVIEW measurements alone to the abstract. This information is contained in the paper, but the way the numbers are presented in the abstract (which is as far as some readers get), it rather underplays the observation constraint brought about by the in-situ measurements alone.

One point that should be added into the discussion: When looking at the ability of a measurement system to assess long-term trends it is critical to consider the length of time over which these measurements are available. In this case, the surface-based network still has an advantage, and does not suffer from the same comparability issues that can arise when new sensors/sampling are introduced. This is mentioned briefly in lines 567-568, but they are first mentioned as a method for satellite validation. Unless this measurements are being made across a profile (such as AirCore or aircraft), I cannot see how this could be the case.

In line 475-476 you mention in passing that your optimisation approach can only solve for constant linear trends over the whole inversions period, which may not be appropriate for China. I wonder if it is really appropriate for other regions either? This is a clear drawback to the choice of state vector in your analytical inversion setup, and should be more clearly stated as such. If you want to test if this lack of trend is consistent with the findings of Sheng et al. (2019), showing an increase to 2012 and a decrease afterwards, perhaps you could perform the same inversion but broken up into two chunks: 2010-2012 and 2013-2017. Yes, this would require new transport simulations, but it would be interesting to check the robustness of the other trends as well. However this might be beyond the scope of the current study. (Perhaps something to add to the discussion?)

I noticed that the panels labelled "China" and "Canada" in Figure 12 are identical. I suspect that they're both showing the results for Canada? In any case, this should be checked carefully and corrected.

Typographical/language remarks:

Co-author Hartmut Boesch's last name is misspelled.

L127: with largest -> with the largest

L162: WETCHART -> WETCHARTS

L169: "full-chemistry" should not be hyphenated here (not a compound adjective before the noun)

L172: closed -> close

L218: challenged -> challenging

L225: Bayesian -> The Bayesian

L231: underestimate -> underestimation

L238: change -> changes

L266: be somewhat deviated -> deviate somewhat; overfit -> overfitting

L278: overfit -> overfitting

L284: Analytical solution -> The analytical solution

L288: I would suggest adding a colon after "analyses"

L290: capitalisation of "In situ-only" seems odd. Perhaps "in-situ-only" would be better as a compound adjective.

L339: year -> years

L345: by year -> by the year

L349: has insignificant -> has an insignificant

L364: higher information than in situ observations -> more information than do in situ observations

L375: I guess that ".," should just be ","?

L392: In situ observation is -> The in situ observations are

L418: Thompton -> Thompson

L453: US -> the US

Figure 11: I guess this percentage change is over the full period (rather than per year)? This should be clarified in the caption label. It also makes it a bit hard to compare to the text, where % trend per year is given. I assume that this is not a compounding percentage change, but rather the total percentage change divided by the number of years? In any case, this should be clarified.

L501-502: This might seem like a small thing, but this is one of the most interesting findings of the paper, and as such should be perfectly clear. I would suggest the following change in phrasing: "are more effective than the satellite observations in independently constraining methane emissions from the sink by OH." -> "are more effective than the satellite observations in constraining methane emissions independently from the OH sink."

L553: weak -> a weak

L560: remove "the"

L561: and methane lifetime -> and a methane lifetime

---

## Referee Comment (RC2) · Anonymous Referee #3 · 20 Nov 2020

"Global methane budget and trend, 2010–2017: complementarity of inverse analyses using in situ (GLOBALVIEWplus CH4 ObsPack) and satellite (GOSAT) observations" presents long-term global inversions based on different available observation datasets. The authors present an inversion system based on the analytical solution of the Bayesian Gaussian problem which allow to better understand the weight of each piece in the system. The authors analyze the outputs thoroughly and use relevant comprehensive metrics to assess the usefulness of each type of observations.

The manuscript is well written, well structured and of significant importance for the

community to be published in ACP after some weaknesses are properly addressed. Main problems are detailed in dedicated sections below and technical revisions are listed in Sect. 5. Overall, the manuscript is of high quality but falls short of properly exploiting the full potential of the system presented here. Sensitivity tests and additional inversions should be added to the manuscript (without computing additional response functions) to prove fully relevant to the community and to stand out of more regular inversion papers. It can be done with relatively little efforts considering all the material and the quality of the background work done to reach the present submitted manuscript.

**1 Bias correction**

p.7 l.191: Bias correction is mentioned. This is a critical point. It may have a huge impact on the inversions. Putting it under the carpet in one line is a little bit short. Please add details on this aspect and possibly some quantification of the impact of such a bias correction. Is the bias correction put in the constant c in eq. (2)? Or is it use on-line in the computation of GEOS-Chem? Or posterior to it? What is the impact on the response functions? If it is the constant c, please include (at least in supplement) your results with/without/with another bias correction to really see how sensitive your results are to that aspect.

**2 Non-linearity of GEOS-Chem and OH chemistry**

This is a little bit harsh to neglect it straight away. Could you run forward runs with your different posterior states and compare with what you get with the matrices Kx to have an idea of how negligible it is?

This may indeed be negligible, but the entire paper is based on that very strong assumption. Please justify it better and more extensively.

**3   Regularization term**

The authors use a regularization term to correct for ill-specified observation errors. However, their estimation is based on approximate matrices. Why not using the rigorous Chi-square criterion? such as in Desroziers et Ivanov (2001, https://rmets. onlinelibrary.wiley.com/doi/10.1002/qj.49712757417)

**4   Computation cost and sensitivity tests**

It is nowhere stated what is the computation cost of the system (computing response functions on the one hand, solving the matrix products on the other hand). Once the response functions are computed it is in principle quite straightforward to change parameters in the R/B matrices to see the impact.

I think the main strength of the system presented here comes from this very fact (other wise, a variational inversion would give posterior fluxes at reduced cost, even if DOFS can be retrieved easily). This is a critical limitation of the present paper.

Different horizontal and temporal correlations should be tested in the prior matrix, as well as standard deviation of errors, to see the impact of such modifications, given that we never really know how good are our prior/obs errors.

More critically are observation errors. Even though the observation data set is very large, it should be possible to imagine a matrix that is diagonal only by block, allowing to consider correlations between GOSAT neighbour observations, while keeping it

possible to compute the inverse easily.

As stated by the authors, the inversions are not consistent with each others (Fig. 13). This comes probably from ill-specified error matrices, which the authors have the tools to inquire into.

**5   Technical comments**

1. p.4 l.89: aircraft measurements: those can be particularly challenging to ingest inversion systems as CTMs never really excel in representing the vertical distribution of CH4 concentrations. Plus it is never clearly stated whether or not they are really used in the inversion or only in the posterior evaluation. Please discuss more about the aircraft measurements and justify better their use (is it only vertical profiles, very hard to assimilate? or transects, easier to use?)

2. p.4 l.104: how exactly the linear trend are computed as response functions? same for OH? A start of explanation is given p.8, but additional information would be welcome

3. p.7 l.163: What is the corresponding total error on the prior budget when using your prior distributed errors? Please represent it on Fig. 13

4. p.8 l.208-213: observation error: it is not clear what ensembles are taken. Do you separate each station? Some regions for GOSAT? etc.

5. p.9 l.284: not correct. The other way around. the analytical solution is the solution of the Bayesian Gaussian problem. The cost function is derived from the formulation of the Gaussian problem when the analytical solution cannot be computed explicitly. Actually, writing the cost function in Eq. (1) in a paper using analytical inversions is superfluous; the factor gamma can be introduced differently

6. p.11 l.376: This warning should also be repeated in the method section. Actually as response functions are computed for each pixels individually, why not duplicating the corresponding time series to separate sectors in the target vector? This would not add new response functions to compute and allow you to assess how good is the distribution in sectors. You could even imagine specifying different correlation lengths to different sectors.

7. p.11 l.382: Is GEOS-Chem really suitable with very coarse resolution to constrain US emissions? the resolution is fine for background sites, but what about sites nearby emission hotspots. Representation errors will likely bias your results at such stations, making it very important to filter properly data prior to the inversion.

---

## Author Comment (AC1) · 26 Dec 2020

**Reviewer #1 Dr. Julia Marshall**

**Comment [1-1]:** This paper presents an analysis of the global methane budget and trend from 2010-2017 by simultaneously optimizing the source distributions, the OH sink (through hemispheric scaling factors), and linear trends using an analytical inversion approach with the GEOS-Chem model. Overall it is clearly written and structured and the figures are sufficiently clear and complete. From the subject matter it fits well within the scope of ACP.

At first glance this paper seems extremely similar in approach and content to Maasakkers et al. (2019) who used a very similar setup with the same model over an overlapping period (2010-2015) to do basically the same thing. The main difference that I can see is that here surface measurements are also included as a data constraint in order to show their complementarity (and consistency).

**Response [1-1]: We thank Dr. Julia Marshall for the positive and valuable comments. All of them have been implemented in the revised manuscript.**

**As mentioned by the reviewer, the main improvement in our work relative to Maasakkers et al. (2019) is adding the in situ observations in the analytical inversion framework, comparing their ability and result with the satellite-based inversion, and quantifying the maximum information from the joint inversion. Such information is extremely important for a better understanding of the methane budgets and for the design of methane observing systems, yet it has not been addressed in previous studies to the best of our knowledge. In addition, our analytical inversion as done here implements a number of improvements to the Maasakkers et al. (2019) methodology, including in particular (1) separate optimization of subcontinental wetland emissions from other emission sectors to resolve their seasonal and interannual variability; (2) optimization of annual hemispheric OH concentrations rather than mean value of the period. Achieving these improvements increases the number of state vectors (and therefore computational costs) by 60%. We believe this work delivers sufficiently novel and important knowledge to the community.**

**Comment [1-2]:** There's something a bit worrying showing up in Figure 6. Figure 6 seems to show that the both the in-situ-only and GOSAT-only inversions overestimate concentrations in the southern hemisphere and underestimate them in the northern hemisphere (more in the mid-latitudes in NH than in the Arctic). Interestingly, this consistent latitude-dependent bias does not seem to be present in the priors, or at least not as strongly. (Note that the 60-90N and 60-90S curves are more or less on top of each other when compared to the observations for the prior runs.) The fact that they then diverge so systematically after optimisation seems to imply that something is going wrong with the OH hemispheric optimisation - or is there another explanation?

Interestingly this pattern appears least distinct when considering the in-situ-only posterior sampled at GOSAT locations, whereas it is most pronounced in the GOSAT-only posterior. Can you explain this? Does this have something to do with the seasonal latitudinal coverage of the GOSAT measurements? In the comparison of the GOSAT-informed concentrations (both with and without the in-situ data) to the ObsPack measurements (panels 6c and 6d, less evident in 6b) there seems to be almost an temporal anti-correlation in the model-data mismatch between the 30-60N stations and the 60-90N stations.

It seems to represent a systematic error in the interhemispheric gradient, which can be explained through either the distribution of the sink, the distribution of the sources, or errors in the transport – or most likely a combination of all three. However as both the sink and the sources are

being optimised, it seems surprising that such a zonally dependent offset is emerging. Even if there are transport errors (and there always are), I would expect a solution to emerge that was consistent with the interhemispheric gradient of the measurements. Of course the OH sink is only being optimised as a hemispheric scaling: might this reflect a problem in the spatial or temporal distribution that is being scaled? Still, usually the fluxes will adapt to compensate, provided they have sufficient flexibility. The fact that Zhang et al. (2018) found the inversion results to be not so sensitive to different OH fields suggests that this is not the case.

Some explanation of the source of this systematic error should be included. The only mention of transport errors is the claim that the regularisation factor gamma should help account for error correlations in the observations due to transport and source aggregation errors. Interestingly this does not seem to appear in the very similar simulations from the same group with a similar set-up, as seen in Figure 3d of Maasakkers et al. (2019).

**Response [1-2]: Thank you very much for pointing this out. We figure out that the hemispheric bias as shown in the original Figure 6 is because the posterior hemispheric OH scaling factors were not correctly implemented in the posterior model simulation. We have corrected the implementation, rerun the posterior model simulation, and updated Figures 5-7 and Tables 1 and 3 in the text. As shown in the updated Figures 6b and 6g, the latitude-dependent bias between the observed and modeled methane concentration has been corrected for the ingested methane observations, indicating that there is no systematic error in the inversion. The updates do not influence the analyses or conclusions. We apologize for the confusion.**

**Figure 6f shows that the in-situ-only inversion biases low to GOSAT observations, and Figure 6c shows that the GOSAT-only inversion overestimates in-situ observations in the Southern Hemisphere while underestimates them in the Northern Hemisphere. These discrepancies, as already presented in the original texts and figure, do not reflect systematic error in the inversion, but rather provide insights on the consistency and complementarity between the two observations in the methane inversion, as analyzed in Section 3.5 and in [Response #1-3]. We have revised the text to clarify.**

**We now state in Section 3.1 "…The in-situ-only inversion effectively corrects this bias and its trend, and also significantly improves the correlations across all platforms. The GOSAT-only inversion performs comparably in correcting the 2010-2017 trend for the independent in-situ data (Fig.6c) and bias for background observations (e.g. aircraft observations in the Southern Hemisphere (Fig.S2)), but there is a low bias at northern mid-latitudes reflecting surface and tower data in North America and Europe. As we will see, the in situ observations are important for optimizing emissions in these regions.**

**… The GOSAT-only inversion corrects the bias and trend in the prior simulation at all latitudes. The in-situ-only inversion corrects the trends, but biases low to the GOSAT observations by about 10 ppbv with larger bias in the Southern Hemisphere due to the sparsity of in situ observation there. The comparison suggests that in situ and GOSAT observations are largely consistent for informing the global methane change but also have some complementarity for the inversion…."**

[Figure]

**Figure 6.** Ability of the inversions to fit the in situ methane observations and GOSAT satellite observations. Panels (a)-(d) show the monthly time series of the differences between observed and simulated in situ methane concentrations averaged over different latitude bands from 2010 to 2017. Panels (e)-(h) are the same as panels (a)-(d) but for GOSAT methane concentrations.

**Comment [1-3]:** Perhaps the most interesting (while also troubling) result is in Figure 13: the negative correlation between methane lifetime and estimated (anthropogenic) emissions is not in and of itself surprising. What is surprising is the fact that none of the three solutions are in any way consistent with each another. This can be explained by an underestimating of the posterior error covariances, as the authors do in L505-509. The fact that the GOSAT+in situ result does not lie somehow between the GOSAT-only and in-situ-only result is, however, worrying. The authors suggest that this is due to a correction of a bias in the GOSAT-only inversion by ingesting the in-situ measurements. This bias was diagnosed as being in both the OH (too low, because the methane in the SH was overestimated) and the fluxes (too low, because the methane in the NH mid-latitudes was underestimated). From this perspective it makes some sense that it would correct in the direction that it did, but why would it overshoot the in-situ-only solution? Is there some fundamental inconsistency in the two types of measurements (or an error in the model) that makes it impossible to match them both simultaneously?

This result seems to suggest that the measurements themselves are not really consistent with each other, which the paper claimed to set out to test (L91-94). Thus this result seems to contradict the conclusion that "the GOSAT and in situ data are generally consistent and can fit each other independently through our inversions" (L535-536). Even if the concentrations in the different inversion come closer to each other, is the result really consistent if the emissions and the lifetime are so very divergent?

**Response [1-3]: The fact that the GOSAT+in situ result does not lie between the GOSAT-only and in-situ-only result (Fig.13) can be inferred from Figures 6c and 6f. Figure 6c suggests that both emissions and OH concentrations are too low in the GOSAT-only inversion, as the reviewer understands, while Figure 6f indicates either underestimation of emissions or overestimation of OH concentrations in the in-situ-only inversion, and the former one is more likely as GOSAT measurements used here are over land which should be more sensitive to emissions than OH loss. The GOSAT + in situ joint inversion thus has to enhance both the**

methane emissions and OH concentrations compared to the In-situ-only and GOSAT-only inversions to correct these biases. We have revised the text accordingly in Section 3.5 to clarify this issue.

We agree with the reviewer that Figure 13 indicates that the measurements are not consistent with each other in optimizing the global methane budget, as stated in the original text (L505-506) "Comparison of the posterior PDFs between the GOSAT-only and In situ-only inversions implies that the two are inconsistent, since the 99% probability contour does not overlap (Fig.13)". We have removed "the GOSAT and in situ data are generally consistent and can fit each other independently through our inversions (L535-536)" which caused confusion. We have revised several places to clarify that the observations are consistent in correcting regional methane emissions in the inversion but are less consistent in terms of informing global methane budgets.

In the abstract, we now state "The in-situ-only and GOSAT-only inversions show consistent corrections to regional methane emissions but are less consistent in optimizing the global methane budget."

In Section 3.5, we now state "Comparison of the posterior PDFs between the GOSAT-only and In-situ-only inversions implies that the two are inconsistent in optimizing global methane budgets, since the 99% probability contours do not overlap (Fig.13a). ... Remarkably, the solution from the GOSAT + in situ joint inversion is more in agreement with in situ observations than GOSAT, and does not lie between these two solutions. Inspection of Figure 6c shows that the GOSAT-only inversion is biased low relative to in situ observations at northern mid-latitudes and biased high in the southern hemisphere, implying that both emissions and OH concentrations are too low. On the other hand, Figure 6f indicates either underestimation of emissions or overestimation of OH concentrations in the in-situ-only inversion, and the former one is more likely as GOSAT measurements used here are over land which should be more sensitive to emissions than OH loss. Ingestion of both observations in the GOSAT + in situ inversion thus enhances both the methane emissions and OH concentrations compared to the in-situ-only and GOSAT-only inversion to correct these biases. It also narrows the posterior error of mean anthropogenic emissions and methane lifetime against tropospheric OH by 20% and 50% compared to the GOSAT-only and in-situ-only inversions, respectively (Fig. 13a). Thus we find that the GOSAT and in situ observations are complementary in quantifying the global budget. "

In the conclusion, we now state "We find that the GOSAT-only inversion can generally fit the in situ data and the in-situ-only inversion can generally fit the GOSAT data, indicating consistency between the two data sets. However, …", "The GOSAT-only and in-situ-only inversions also show consistent corrections to regional methane emissions in the US, Europe, and China.", and "GOSAT and in situ observations have complementarity in constraining global emissions."

**Comments [1-4]:** While trying to understand this rather surprising result I realised that I would like to see some more figures: OH was scaled per hemisphere per year (16 state vector values). A time series of these scaling factors (perhaps as an additional panel or two in Figure 7?) would be interesting to see, rather than just an average lifetime over the whole period (similar to Figure 7d in Maasakkers et al. (2019)). This might also help convince me that scaling OH based on surface-based

methane measurements alone makes sense - do the OH scaling factors in this case stay close to one throughout?

Another plot that might help convince the reader of the adequacy of the transport model and the improvement of the sources and sinks would be geographical (zonal + altitude?) plot of the model-data mismatch for aircraft data presented in Figure 5d. Even if it has to go into a supplement, it would be a useful piece of information for the reader to assess if this very surprising result might make sense.

Once these concerns are addressed, I think the paper would be appropriate for publication in ACP.

**Response [1-4]: Thank you for the advice, we have added the two figures (Fig.7b and Fig.S2) and revised the text accordingly.**

**1) We present the posterior methane lifetime (as an indicator of OH scaling factors) in Figure 7b. We now state in Section 3.5 "We also find that the in-situ-only inversion yields a larger interannual variability of posterior OH concentrations and thus methane lifetime than the GOSAT-only inversion (Fig.7b), due to the heterogeneous spatial and temporal distribution of the in situ observations.".**

[Figure]

**Figure 7.** (a) Annual global growth rate of atmospheric methane, 2010-2017. Results from our three different inversions (In-situ-only, GOSAT-only, GOSAT + in situ) are compared to the observed growth rates inferred from the NOAA surface observational network (https://www.esrl.noaa.gov/gmd/ccgg/trends_ch4/, last access: 20 June, 2020). Mean annual growth rates and standard deviations from the different inversions are shown inset. (b). Methane lifetime against oxidation by tropospheric OH, 2010-2017, from the three different inversions. Mean lifetime and standard deviations are shown inset. The methane lifetime in the prior estimate is 10.6 years.

**2) We present the model-observation bias for aircraft data for the prior and posterior**

**simulation in Fig.S2, and state in Section 3.1 "The GOSAT-only inversion performs comparably in correcting the 2010-2017 trend for the independent in-situ data (Fig.6c) and bias for background observations (e.g. aircraft observations in the Southern Hemisphere (Fig.S2))"**

[Figure]

**Figure S2.** Differences between simulated and observed aircraft methane concentrations from the GLOBALVIEWplus ObsPack data product using GEOS-Chem with prior estimates and with posterior estimates from the in-situ-only, GOSAT-only, and GOSAT + in situ inversions.

**Comments [1-5]:** Minor comments: I would recommend adding how many independent pieces of information are contained in the GLOBALVIEW measurements alone to the abstract. This information is contained in the paper, but the way the numbers are presented in the abstract (which is as far as some readers get), it rather underplays the observation constraint brought about by the in-situ measurements alone.

**Response [1-5]: We have revised accordingly in the abstract "The in-situ-only and the GOSAT-only inversion alone, achieve respectively 113 and 212 independent pieces of information (DOFS) for quantifying mean 2010-2017 anthropogenic emissions on 1009 global model grid elements, and DOFS of 67 and 122 for 2010-2017 emission trends. The joint GOSAT + in situ inversion achieves DOFS of 262 and 161 respectively for mean emissions and trends. The in situ data thus increase the global information content from the GOSAT-only inversion by 20-30%."**

**Comments [1-6]:** One point that should be added into the discussion: When looking at the ability of a measurement system to assess long-term trends it is critical to consider the length of time over which these measurements are available. In this case, the surface-based network still has an advantage, and does not suffer from the same comparability issues that can arise when new sensors/sampling are introduced. This is mentioned briefly in lines 567-568, but they are first

mentioned as a method for satellite validation. Unless this measurements are being made across a profile (such as AirCore or aircraft), I cannot see how this could be the case.

**Response [1-6]: We agree. We now rephrased in the Section 3.5 "In situ observations will in any case continue to play a critical role for documenting long-term trends of methane with consistent calibration, …".**

**Comments [1-7]:** In line 475-476 you mention in passing that your optimisation approach can only solve for constant linear trends over the whole inversions period, which may not be appropriate for China. I wonder if it is really appropriate for other regions either? This is a clear drawback to the choice of state vector in your analytical inversion setup, and should be more clearly stated as such. If you want to test if this lack of trend is consistent with the findings of Sheng et al. (2019), showing an increase to 2012 and a decrease afterwards, perhaps you could perform the same inversion but broken up into two chunks: 2010-2012 and 2013-2017. Yes, this would require new transport simulations, but it would be interesting to check the robustness of the other trends as well. However this might be beyond the scope of the current study. (Perhaps something to add to the discussion?)

**Response [1-7]: We agree, and indeed separating the inversions into two or more chunks will increase significantly the computational costs. We have clarified this limitation in Section 2.2: "The inclusion of linear trends in state vectors allows us to identify the direction of emission change for each 4° ×5° grid in the 8-year period, but it would not capture high-frequency interannual variability."**

**Comments [1-8]:** I noticed that the panels labelled "China" and "Canada" in Figure 12 are identical. I suspect that they're both showing the results for Canada? In any case, this should be checked carefully and corrected.

**Response [1-8]: Thanks for pointing it out. We had corrected the figure before it was posted on ACP Discussion.**

Typographical/language remarks:

**Comments [1-8]:** Co-author Hartmut Boesch's last name is misspelled.

**Response[1-8]: Corrected**

**Comments [1-9]:**L127: with largest -> with the largest

**Response[1-9]: Corrected**

**Comments [1-10]:**L162: WETCHART -> WETCHARTS

**Response[1-10]: Corrected**

**Comments [1-11]:**L169: "full-chemistry" should not be hyphenated here (not a compound adjective before the noun)

**Response[1-11]: Corrected**

**Comments [1-12]:**L172: closed -> close

**Response[1-12]: It has been rephrased.**

**Comments [1-13]:**L218: challenged -> challenging
**Response[1-13]: Corrected**

**Comments [1-14]:**L225: Bayesian -> The Bayesian
**Response[1-14]: Corrected**

**Comments [1-15]:**L231: underestimate -> underestimation
**Response[1-15]: Corrected**

**Comments [1-16]:**L238: change -> changes
**Response[1-16]: Corrected**

**Comments [1-17]:**L266: be somewhat deviated -> deviate somewhat; overfit -> overfitting
**Response[1-17]: It has been removed.**

**Comments [1-18]:**L278: overfit -> overfitting
**Response[1-18]: Corrected**

**Comments [1-19]:**L284: Analytical solution -> The analytical solution
**Response[1-19]: Corrected**

**Comments [1-20]:**L288: I would suggest adding a colon after "analyses"
**Response[1-20]: Corrected**

**Comments [1-21]:**L290: capitalisation of "In situ-only" seems odd. Perhaps "in-situ-only" would
be better as a compound adjective.
**Response[1-21]: Corrected**

**Comments [1-22]:**L339: year -> years
**Response[1-22]: Corrected**

**Comments [1-23]:**L345: by year -> by the year
**Response[1-23]: Corrected**

**Comments [1-24]:**L349: has insignificant -> has an insignificant
**Response[1-24]: Corrected**

**Comments [1-25]:**L364: higher information than in situ observations -> more information than do
in situ observations
**Response[1-25]: Corrected**

**Comments [1-26]:**L375: I guess that ".," should just be ","?
**Response[1-26]: Corrected**

**Comments [1-27]:**L392: In situ observation is -> The in situ observations are
**Response[1-27]: Corrected**

**Comments [1-28]:**L418: Thompton -> Thompson
**Response[1-28]: Corrected**

**Comments [1-29]:**L453: US -> the US
**Response[1-29]: Corrected**

**Comments [1-30]:** Figure 11: I guess this percentage change is over the full period (rather than per year)? This should be clarified in the caption label. It also makes it a bit hard to compare to the text, where % trend per year is given. I assume that this is not a compounding percentage change, but rather the total percentage change divided by the number of years? In any case, this should be clarified.
**Response [1-30]: Figure 11 shows the percentage change per year that derive directly from the inversions. We now state in the figure caption "Figure 11. Same as Figure 8 but for optimization of non-wetland (mainly anthropogenic) emission trends (% a$^{-1}$) in 2010-2017.".**

**Comments [1-31]:**L501-502: This might seem like a small thing, but this is one of the most interesting findings of the paper, and as such should be perfectly clear. I would suggest the following change in phrasing: "are more effective than the satellite observations in independently constraining methane emissions from the sink by OH." -> "are more effective than the satellite observations in constraining methane emissions independently from the OH sink."
**Response [1-31]: We have rephrased as suggested.**

**Comments [1-32]:**L553: weak -> a weak
**Response[1-32]: Corrected**

**Comments [1-33]:**L560: remove "the"
**Response[1-33]: Corrected**

**Comments [1-34]:**L561: and methane lifetime -> and a methane lifetime
**Response[1-34]: Corrected**

---

## Author Comment (AC2) · 26 Dec 2020

**Reviewer #2**

**Comments [2-1]:** "Global methane budget and trend, 2010–2017: complementarity of inverse analyses using in situ (GLOBALVIEWplus CH4 ObsPack) and satellite (GOSAT) observations" presents long-term global inversions based on different available observation datasets. The authors present an inversion system based on the analytical solution of the Bayesian Gaussian problem which allow to better understand the weight of each piece in the system. The authors analyze the outputs thoroughly and use relevant comprehensive metrics to assess the usefulness of each type of observations.

The manuscript is well written, well structured and of significant importance for the community to be published in ACP after some weaknesses are properly addressed. Main problems are detailed in dedicated sections below and technical revisions are listed in Sect. 5. Overall, the manuscript is of high quality but falls short of properly exploiting the full potential of the system presented here. Sensitivity tests and additional inversions should be added to the manuscript (without computing additional response functions) to prove fully relevant to the community and to stand out of more regular inversion papers. It can be done with relatively little efforts considering all the material and the quality of the background work done to reach the present submitted manuscript.

**Response [2-1]: We thank the reviewer for the positive and valuable comments. All of them have been implemented in the revised manuscript. In particular, we have performed a number of additional inversions to test the sensitivity of our results to the choices in cost-function construction (e.g. usage of observations, error assumption of the observations and state). Please see our itemized responses below.**

**Comments [2-2]:** 1 Bias correction: p.7 l.191: Bias correction is mentioned. This is a critical point. It may have a huge impact on the inversions. Putting it under the carpet in one line is a little bit short. Please add details on this aspect and possibly some quantification of the impact of such a bias correction. Is the bias correction put in the constant c in eq. (2)? Or is it use on-line in the computation of GEOS-Chem? Or posterior to it? What is the impact on the response functions? If it is the constant c, please include (at least in supplement) your results with/without/with another bias correction to really see how sensitive your results are to that aspect.

**Response [2-2]: Thanks for pointing it out. The bias correction is done off-line before the inversion. We have added the text briefly describing the procedures for bias correction, and a Figure S1 to show the influence of bias correction. We now state in Section 2.3 "GEOS-Chem has excessive methane in the high-latitudes stratosphere, a flaw common to many models (Patra et al., 2011) especially at coarse model resolution. Following Zhang et al. (2020), we compute correction factors to GEOS-Chem stratospheric methane subcolumns as a function of season and equivalent latitude to match the measurements from the solar occultation ACE-FTS v3.6 instrument (Waymark et al., 2014; Koo et al., 2017). As shown in Zhang et al. (2020), the correction can be up to 10% at high latitudes during winter and spring. We apply the correction factors before the inversion to avoid wrongly attributing this model transport bias to methane emissions and loss. Figure S1 shows that the systematic differences in the posterior scaling factors of non-wetland emissions with vs. without bias correction are more prominent at the northern high latitudes, as also shown in Stanevich et al. (2020), but the global total emissions only differ by 1%. "**

[Figure]

**Figure S1.** Posterior scaling factors of non-wetland methane emissions from GOSAT-only inversion (a) with GOSAT stratospheric bias corrections and (b) without GOSAT stratospheric bias corrections.

**Reference:**

Stanevich, I., Jones, D. B. A., Strong, K., Parker, R. J., Boesch, H., Wunch, D., Notholt, J., Petri, C., Warneke, T., Sussmann, R., Schneider, M., Hase, F., Kivi, R., Deutscher, N. M., Velazco, V. A., Walker, K. A., and Deng, F.: Characterizing model errors in chemical transport modeling of methane: impact of model resolution in versions v9-02 of GEOS-Chem and v35j of its adjoint model, Geosci. Model Dev., 13, 3839–3862, https://doi.org/10.5194/gmd-13-3839-2020, 2020.

Zhang, Y., Jacob, D. J., Lu, X., Maasakkers, J. D., Scarpelli, T. R., Sheng, J.-X., Shen, L., Qu, Z., Sulprizio, M. P., Chang, J., Bloom, A. A., Ma, S., Worden, J., Parker, R. J., and Boesch, H.: Attribution of the accelerating increase in atmospheric methane during 2010–2018 by inverse analysis of GOSAT observations, Atmos. Chem. Phys. Discuss., https://doi.org/10.5194/acp-2020-964, in review, 2020.

**Comments [2-3]:** 2 Non-linearity of GEOS-Chem and OH chemistry. This is a little bit harsh to neglect it straight away. Could you run forward runs with your different posterior states and compare with what you get with the matrices Kx to have an idea of how negligible it is?

**Response [2-3]: The GEOS-Chem methane simulation used prescribed monthly 3-D fields of global tropospheric OH concentrations taken from a GEOS-Chem simulation with full chemistry. With this regard the optimization of methane emissions is strictly linear. The only non-linearity emerges regarding the optimization of OH, because the sensitivity of the methane concentration to changes in OH concentrations depends on the methane concentration through first-order loss, but the variability of methane concentration is sufficiently small so that this non-linearity is negligible. We have tested that the $K\hat{x}$ and posterior simulation of y has a small mean difference of 2±3 ppbv. We now state in Section 2.4** **"The optimization of methane emission and its trends is strictly linear by design because we use prescribed monthly 3-D OH fields as described in Section 2.2. There is some non-linearity regarding the optimization of OH, because the sensitivity of the methane concentration to changes in OH concentrations depends on the methane concentration through first-order loss,**

**but we assume that the variability of methane concentration is sufficiently small that this non-linearity is negligible (we verify this assumption below)…. Comparison of the resulting Jacobian matrix to GEOS-Chem as $F(x) - Kx - c$ shows a negligible residual difference of $2\pm3$ ppb, verifying the assumption of linearity."**

**Comments [2-4]:** 3 Regularization term: The authors use a regularization term to correct for ill-specified observation errors. However, their estimation is based on approximate matrices. Why not using the rigorous Chi-square criterion? such as in Desroziers et Ivanov (2001, https://rmets. onlinelibrary.wiley.com/doi/10.1002/qj.49712757417)

**Response [2-4]: Thanks for pointing it out. We have made the revision to estimate the optimal value of the regularization parameter in the context of the Chi-square distribution. We have also tested the impact of using different regularization parameters on the global methane budget as discussed in [Response #2-5].**

**We now state in Section 2.4 "… For a given state vector element $i$, the expected value of $(x_i - x_{Ai})^2$ is the prior error variance $\sigma_{Ai}^2$. For an $n$-dimensional state vector with a diagonal prior error covariance matrix, the state component $J_A$ of the cost function is the sum of $n$ random normal elements**

$$J_A(x) = (x - x_A)^T S_A^{-1}(x - x_A) = \sum_n \frac{(x_i - x_{Ai})^2}{\sigma_{Ai}^2} \text{ (6)},$$

**and its pdf is given by the Chi-square distribution with $n$ degrees of freedom ($n$=3378 in this case), with an expected value of $n$ and a standard deviation of $\sqrt{2n}$. One can apply the same reasoning to the observation component $J_O$ of the posterior cost function,**

$$J_O(x) = (y - Kx)^T S_O^{-1}(y - Kx) = \sum_m \frac{(y_i - Kx_i)^2}{\sigma_{oi}^2} \text{ (7)},$$

**whose pdf follows a chi-square distribution with $m$ degrees of freedom. However, this component is less sensitive to the choice of $\gamma$ because of the large random error component for individual observations.**

**Figure 4 shows the dependences of $J_A(\hat{x})$ and $J_O(\hat{x})$ on the choice of the regularization parameter $\gamma$, for the in situ and GOSAT observations. The in situ observations are sufficiently sparse that $\gamma = 1$ (no regularization) is expected. In the case of GOSAT, however, $\gamma = 1$ would yield $J_A(\hat{x}) = 6n \gg n + \sqrt{2n}$ which indicates overfitting, while $\gamma = 0.1$ yields $J_A(\hat{x}) \approx n$ which is the expected value and is used here…."**

**Comments [2-5]:** 4 Computation cost and sensitivity tests. It is nowhere stated what is the computation cost of the system (computing response functions on the one hand, solving the matrix products on the other hand). Once the response functions are computed it is in principle quite straightforward to change parameters in the R/B matrices to see the impact.

I think the main strength of the system presented here comes from this very fact (otherwise, a variational inversion would give posterior fluxes at reduced cost, even if DOFS can be retrieved easily). This is a critical limitation of the present paper.

Different horizontal and temporal correlations should be tested in the prior matrix, as well as standard deviation of errors, to see the impact of such modifications, given that we never really know how good are our prior/obs errors.

More critically are observation errors. Even though the observation data set is very large, it should be possible to imagine a matrix that is diagonal only by block, allowing to consider correlations between GOSAT neighbour observations, while keeping it possible to compute the inverse easily. As stated by the authors, the inversions are not consistent with each others (Fig. 13). This comes probably from ill-specified error matrices, which the authors have the tools to inquire into.

**Response [2-5]: Thank you for pointing it out.**

**1) We have added the following text in Section 2.4 (Analytical Inversion) to clarify the computation cost of the system "A requirement of the analytical approach is that the Jacobian matrix be explicitly constructed, requiring $n + 1$ forward model runs. Building the Jacobian matrix for the 3378 state vectors in this 8-year period study requires about one million core hours (8 cores × 36 hours per simulation × 3378 simulations). However, this construction is readily done in parallel on high-performance computing clusters.".**

**2) We have also conducted a number of additional inversions to examine the results with different error assumption and ingestion of observations. For the prior standard deviation of state vectors (non-wetland emission trends and OH), we test their different magnitude (decrease by 50%) but not their distributions (correlations) due to the lack of objective information on the later. For the observation error, the ability to test off-diagonal assumption is also limited by the calculation of $S_O{}^{-1}$ which involves inverting a matrix with ~$10^{12}$ elements. Therefore we test the unknown observation error correlations by changing the regularization parameter $\gamma$.**

**We have added a new Figure 13b, and now state in Section 2.4 "We will make use of these advantages in comparing the ability of the in-situ-only, GOSAT-only, and GOSAT + in situ inversions, and to test how choices in cost-function construction affect our conclusions including changing the regularization parameter $\gamma$, changing the prior error estimates, and using different types of in-situ observations. Our analysis will focus on results from the base inversions with the default settings, but we will use results from the sensitivity inversions to address specific issues.".**

**And in Section 3.5 we state "We examine in Figure 13b the sensitivity of the global methane budget optimization to the choice of different regularization parameter $\gamma$ (and therefore observation error $S_O$) and prior error of methane emission trends and OH concentrations. We find that reducing $\gamma$ or prior errors of trend and OH by 50% yields consistent estimates of anthropogenic emissions and OH concentrations as compared to the default inversion, with differences within 3%. Decreasing the weighting of observations in the inversion (i.e. assuming larger observation error) enlarges the posterior error and pushes the posterior estimates closer to the prior estimates. Assuming a lower prior error for OH concentration from 10% to 5% results in lower methane lifetime (closer to the prior) and higher emissions, and also reduces the error correlation between the optimization of methane emissions and OH, while assuming a lower prior error for non-wetland emission trends leads to an opposite effect. Our results are consistent with Maasakkers et al. (2019), which shows that different assumptions of error distribution and magnitude tin their analyses have relatively small results. We also find that having the shipboard and aircraft measurements in the in-situ-only inversion pushes the estimate to be more consistent with the GOSAT-only**

**inversion (Fig.13b), implying that the shipboard and aircraft measurements by emphasizing the methane in the remote atmosphere play a similar role as satellite measurements in global methane budget optimization."**

[Figure]

**Figure 13.** Joint probability density functions (PDFs) of global mean anthropogenic methane emission and methane lifetime against oxidation by tropospheric OH optimized by different inversions. Panel (a) shows the results from the prior and the three base inversions. The prior estimates are shown in grey with bars representing the prior error standard deviation. The thick contours show probabilities of 0.99 (outermost), 0.7, 0.5, 0.3, and 0.1 (innermost). The error correlation coefficients are given inset. Panel (b) shows the 0.99 probability contours from the three base inversions along with the same contours for ten additional sensitivity inversions using reduced values of the regularization parameter $\gamma$ (0.05 instead of 0.1 for GOSAT, 0.5 instead of 1 for in situ); reduced errors for the methane emission trends on the $4°\times5°$ grid (5% $a^{-1}$ instead of 10% $a^{-1}$); reduced errors on annual hemispheric mean OH concentrations (5% instead of 10%); or surface and tower data only in the in-situ-only inversion.

**Comments [2-6]:** 5 Technical comments. p.4 l.89: aircraft measurements: those can be particularly challenging to ingest inversion systems as CTMs never really excel in representing the vertical distribution of CH4 concentrations. Plus it is never clearly stated whether or not they are really used in the inversion or only in the posterior evaluation. Please discuss more about the aircraft measurements and justify better their use (is it only vertical profiles, very hard to assimilate? or transects, easier to use?)

**Response [2-6]: Thank you for pointing it out.**

**1) The aircraft measurements are used in the inversions, as stated in the original text (L122-124) "We obtain in this manner 157054 observation data points for the inversion including 81119 from 103 surface sites, 27433 from 13 towers, 827 from 3 ship cruises, and 47675 from 29 aircraft campaigns.". We have added a Figure S2 to also address [Comment #1-4], which shows that the posterior model can well fit the aircraft methane measurements measuring the background (e.g. in the Southern Hemisphere), but indeed some discrepancies emerge in the northern mid-latitudes, reflecting the difficulty in modeling methane vertical distributions or optimizing emissions near source.**

**2) We have also added an additional inversion using only surface and tower observations in the inversion and compared the results with the In-situ-only inversion (which ingest all in situ observations) in Fig.S3 and Fig.13b. Comparison of Figure S3 to Figure 8a-b shows that**

adding the aircraft and shipboard observations to the surface and tower observations increases the DOFS for constraining non-wetland methane emissions from 96 to 113 (18%), and reflects the upward correction in the South America which is consistent with the GOSAT-only inversion (Fig.8d). We also find in the Figure 13b that adding the aircraft and shipboard measurements pushes the inversed global methane and OH levels more consistent with the GOSAT-only inversion, however, it makes the inversion less effective in optimizing the global methane budget and OH. These results thus illustrate the ability of aircraft and shipboard measurements in the inversion.

We now state in Section 3.2 "We find that the DOFS from the in-situ-only inversion observations are mostly (85%) from the surface and tower measurements (Fig.S3)."

We also state in Section 3.5 "…A sensitivity inversion using only the surface and tower measurements in the In-situ-only inversion yields $r$=-0.37 (Fig.13b). It indicates that in situ observations, in particular surface and tower measurements, are more effective than the satellite observations in constraining methane emissions independently from the sink by OH.", and "We also find that having the shipboard and aircraft measurements in the in-situ-only inversion pushes the estimate to be more consistent with the GOSAT-only inversion (Fig.13b), implying that the shipboard and aircraft measurements by emphasizing the methane in the remote atmosphere play a similar role as satellite measurements in global methane budget optimization."

[Figure]

**Figure S3**. Same as Figure 8a and 8b but from a sensitivity inversion using only surface and tower methane observations.

**Comments [2-7]:** p.4 l.104: how exactly the linear trend are computed as response functions? same for OH? A start of explanation is given p.8, but additional information would be welcome.

**Response [2-7]: We now state in the text to introduce the construction of response functions (Jacobian matrix $K$) in Section 2.4: "We construct the Jacobian matrix $K$ explicitly by conducting GEOS-Chem simulations with each element of the state vector perturbed separately. For the linear emission trend elements, this is done by perturbing the 2010-2017 emission trend in each grid cell from 0% (the best prior estimate) to 10% $a^{-1}$; for OH, this is done by perturbing yearly hemispheric OH fields by 20% without modifying the spatial or seasonal distribution."**

**Comments [2-8]:** p.7 l.163: What is the corresponding total error on the prior budget when using your prior distributed errors? Please represent it on Fig. 13

**Response [2-8]: We have revised Fig.13 accordingly.**

**Comments [2-9]:** p.8 l.208-213: observation error: it is not clear what ensembles are taken. Do you separate each station? Some regions for GOSAT? etc.

**Response [2-9]: We now state in Section 2.3: "For in-situ observations, we derive $\varepsilon_0$ separately for the ensemble of background surface sites (Dlugokencky et al., 1994), non-background sites, tower sites, shipboard measurements, and aircraft measurements, while for GOSAT observations $\varepsilon_0$ is calculated for each $4° \times 5°$ grid cell."**

**Reference**

Dlugokencky, E. J., Steele, L. P., Lang, P. M., and Masarie, K. A.: The growth rate and distribution of atmospheric methane, J. Geophys. Res., 99, 17021, http://doi.org/10.1029/94jd01245, 1994.

**Comments [2-10]:** p.9 l.284: not correct. The other way around. the analytical solution is the solution of the Bayesian Gaussian problem. The cost function is derived from the formulation of the Gaussian problem when the analytical solution cannot be computed explicitly. Actually, writing the cost function in Eq. (1) in a paper using analytical inversions is superfluous; the factor gamma can be introduced differently.

**Response [2-10]: We have rephrased as "The analytical solution to the Bayesian optimization problem, as done here, has several advantages relative to the more commonly used variational (numerical) solution."**

**Comments [2-11]:** p.11 l.376: This warning should also be repeated in the method section. Actually as response functions are computed for each pixels individually, why not duplicating the corresponding time series to separate sectors in the target vector? This would not add new response functions to compute and allow you to assess how good is the distribution in sectors. You could even imagine specifying different correlation lengths to different sectors.

**Response [2-11]: We cannot separate sectors at the level of individual grid cells because they will all have the same response function. We can separate sectors for ensembles of grid cells and this is precisely what we do with the matrix *W*. We have added the following text in Section 2.4 "We cannot separate individual sectors within a $4° \times 5°$ grid cell because they will all have the same response function (Jacobian column). However, we can aggregate results spatially and by sector…"**

**Comments [2-12]:** p.11 l.382: Is GEOS-Chem really suitable with very coarse resolution to constrain US emissions? the resolution is fine for background sites, but what about sites nearby emission hotspots. Representation errors will likely bias your results at such stations, making it very important to filter properly data prior to the inversion.

**Response [2-12]: Thanks for pointing it out. We agree that representation errors will likely bias results at stations near source regions, and it is important to filter properly data prior to the inversion. As already mentioned in Section 2.1, we address this problem by "For surface and tower measurements, we use only daytime (10-16 local time) observations and average them to the corresponding daytime mean values. We exclude outliers at individual sites that depart by more than three standard deviations from the mean.". Still this might be insufficient**

to properly interpret sites nearby emission hotspots. A high-resolution inversion (e.g. Turner et al., 2015; Sheng et al., 2018) would be preferable to better interpret the in-situ observations near emission hotspots and to understand the spatial pattern of US anthropogenic methane emissions.

**Reference:**

Sheng, J.-X., Jacob, D. J., Turner, A. J., Maasakkers, J. D., Sulprizio, M. P., Bloom, A. A., Andrews, A. E., and Wunch, D.: High-resolution inversion of methane emissions in the Southeast US using SEAC4RS aircraft observations of atmospheric methane: anthropogenic and wetland sources, Atmos. Chem. Phys., 18, 6483-6491, http://doi.org/10.5194/acp-18-6483-2018, 2018.

Turner, A. J., Jacob, D. J., Wecht, K. J., Maasakkers, J. D., Lundgren, E., Andrews, A. E., Biraud, S. C., Boesch, H., Bowman, K. W., Deutscher, N. M., Dubey, M. K., Griffith, D. W. T., Hase, F., Kuze, A., Notholt, J., Ohyama, H., Parker, R., Payne, V. H., Sussmann, R., Sweeney, C., Velazco, V. A., Warneke, T., Wennberg, P. O., and Wunch, D.: Estimating global and North American methane emissions with high spatial resolution using GOSAT satellite data, Atmos. Chem. Phys., 15, 7049-7069, http://doi.org/10.5194/acp-15-7049-2015, 2015.

---

## Author Comment (AC3) · 26 Dec 2020

**Response to the short comment on "Global methane budget and trend, 2010-2017: complementarity of inverse analyses using in situ (GLOBALVIEWplus CH4 ObsPack) and satellite (GOSAT) observations" by Dr. Luke Western**

*Xiao Lu, Daniel Jacob, Yuzhong Zhang, on behalf of co-authors*
*Harvard John A. Paulson School of Engineering and Applied Sciences, Harvard University, Cambridge, MA, USA*

We appreciate Dr. Luke Western's attention and comments (hereafter Western (2020)) on our work (hereafter Lu et al. (2020)). We organize our response as 1) clarification of some misinterpretation of Lu et al. (2020) in Western (2020), 2) response to the need of parameterization parameter $\gamma$ (rather than using other methods) and its role in the Bayesian inversion framework (respond to Section 3 in Western (2020)), and 3) response to the concerns of using $J_A(\hat{x}) \approx n$ and $J_O(\hat{x}) \approx m$ to inform the optimal parameterization parameter $\gamma$ (respond to Section 2 in Western (2020)).

1). We believe that Western (2020) at least partly misinterprets the statements in Lu et al. (2020). Western (2020) states in Page 3, the end of Section 2 that "If $J_A(\hat{x}) < n$, and $J_O(\hat{x}) < m$, why would this suggest an overfit?" and in Page 3, the beginning of Section 3 "My interpretation of, for example, equation 2 (Lu et al., 2020, equation 7), is that if $J_O(\hat{x}) \leq m$ in equation 2, one would assume that the inversion is over confident in its estimated value, and hence the uncertainty is smaller than it should be.". Our statement was "Nevertheless, $J_A(\hat{x}) \gg n$ implies overfit to the observations because the posterior state vector estimates are far outside the estimated errors on the prior estimates." (Line 266). This is not contradict to Western (2020).

2). The need for regularization parameter γ is to avoid overfitting to the observations because the number of observations (1.6 million for GOSAT) is much larger than the number of state vector elements (3378), and the error covariance of the observations cannot be properly quantified. We agree with Western (2020) that there are alternatives to adjust probability distributions rather than using regularization parameter, but they are not really practical in our case. The first alternative mentioned in Western (2020) is to explicitly include uncertainty in parameters within the inversion itself following either an empirical Bayes or hierarchical approach. However, this method does not provide explicitly the analytical solution (which is critical to our objective to quantify the information from inversion) and requires additional analyses, e.g. applying Markov chain Monte Carlo (MCMC) (Ganesan et al., 2014). The second is to create a better prior probability distribution that is representative of actual prior. This is mainly limited by the fact that we do not actually have objective information of the covariance for both state vectors and observations. Therefore, we argue that applying the regularization parameter is the simplest and most applicable method in our inversion framework. The regularization factor itself does not alter the Bayesian nature of the inversion--it is intended to address the lack of covariance structure in the error covariance matrices by modifying the weighting of the prior and observational terms.

3). We agree with Western (2020) that the $J_A(x)$ (similarly $J_O(x)$) should be expressed in the context of Chi-square distribution with *n* degrees of freedom (*n*=3378 in this case), an expected value of *n* ($E(J_A(x) = n)$), and a standard deviation of $\sqrt{2n}$. Note that the Chi-square distribution

converges toward a normal distribution for large degrees of freedom (which is the case here). Interpretation of Figure 4 in Lu et al. (2020) shows that the solution with $\gamma = 1$ for the GOSAT-only inversion yields the $J_A(\hat{x}) = 6n \gg n \pm \sqrt{2n}$, suggesting that the posterior state vector estimates are far outside the estimated errors on the prior estimate which indicates overfitting. Our application of comparing $J_A(\hat{x})$ to $n$ for overfitting checking yields consistent result with Zhang et al. (2018), which used the L-curve method (Hansen, 2000) to determine the optimal regularization parameter $\gamma$ to be 0.05-0.1 for GOSAT observation in a global inversion at $4° \times 5°$ resolution. This method also provides a way to properly weigh the in situ and GOSAT observations in the inversion, by comparing $J_A(\hat{x})$ from the two inversions as shown in the Figure 4. We have revised the text accordingly.

In summary, we appreciate that Dr. Luke Western raises the discussion on this issue and helps to improve the clarification and presentation of the methods. Our method by applying a regularization parameter provides a means to account for unknown error covariance and to weigh different observations in the inversion, while we agree there are alternatives that can be more rigorous and advantageous in some cases. This will be an open question and we welcome further discussions and studies on this issue.

**Reference:**

Ganesan, A. L., Rigby, M., Zammit-Mangion, A., Manning, A. J., Prinn, R. G., Fraser, P. J., Harth, C. M., Kim, K.-R., Krummel, P. B., Li, S., Mühle, J., O'Doherty, S. J., Park, S., Salameh, P. K., Steele, L. P., and Weiss, R. F.: Characterization of uncertainties in atmospheric trace gas inversions using hierarchical Bayesian methods, Atmos. Chem. Phys., 14, 3855–3864, https://doi.org/10.5194/acp-14-3855-2014, 2014.

Hansen, P. C.: The L-Curve and its Use in the Numerical Treatment of Inverse Problems, in: Computational Inverse Problems in Electrocardiology, edited by: Johnston, P., Advances in Computational Bioengineering, WIT Press, 119–142, 2000.

Lu, X., Jacob, D. J., Zhang, Y., Maasakkers, J. D., Sulprizio, M. P., Shen, L., Qu, Z., Scarpelli, T. R., Nesser, H., Yantosca, R. M., Sheng, J., Andrews, A., Parker, R. J., Boech, H., Bloom, A. A., and Ma, S.: Global methane budget and trend, 2010–2017: complementarity of inverse analyses using in situ (GLOBALVIEWplus CH$_4$ ObsPack) and satellite (GOSAT) observations, Atmos. Chem. Phys. Discuss., https://doi.org/10.5194/acp-2020-775, in review, 2020.

Western, Luke: Short comment to Lu et al.: Global methane budget and trend, 2010–2017: complementarity of inverse analyses using in situ (GLOBALVIEWplus CH4ObsPack) and satellite (GOSAT) observation'

Zhang, Y., Jacob, D. J., Maasakkers, J. D., Sulprizio, M. P., Sheng, J.-X., Gautam, R., and Worden, J.: Monitoring global tropospheric OH concentrations using satellite observations of atmospheric methane, Atmos. Chem. Phys., 18, 15959-15973, http://doi.org/10.5194/acp-18-15959-2018, 2018.

---

## Author Response (AR2)

Dear Dr. Patrick Jöckel,

**Thank you very much for handling our manuscript. Please find below our itemized responses to the reviewer's comments.**

**Thank you for your consideration.**

**Sincerely,**
**Xiao Lu et al.**
* * *
**Reviewer #1 Dr. Julia Marshall**

**Comment [1-1]:** The paper is much improved after revision! I have only a couple very minor comments about the new material that was added, which might clarify the interpretation. The following paragraph remains a sticking point for me:

In L575-L579 of the revised manuscript the author's state: "...in situ observations, in particular surface and tower measurements, are more effective than the satellite observations in independently constraining methane emissions independently from the sink by OH", but also that "the in-situ-only inversion yields a larger interannual variability of posterior OH concentrations and thus methane lifetime than the GOSAT only inversion (Fig.7b), due to the heterogeneous spatial and temporal distribution of the in situ observations."

Both of these things cannot be true. It cannot be that the in-situ measurements better constrain the emissions independently of the OH fields while simultaneously leading to more OH interannual variability. I think the latter finding (more interannual variability in OH in the surface-only inversion) is instead related to having only trends to optimize in anthropogenic emissions. If the model cannot allow emissions to e.g. first decrease and then increase, it can only match the data by adjusting the sink! But this does not necessarily make it a physically reasonable solution. I think this needs to be discussed in some more detail.

**Response [1-1]: Thanks for pointing it out. The larger interannual variability of posterior OH in the in-situ-only inversion is mostly because the number and location of in situ observations varies in different years, in particularly for aircraft and shipboard observations. Our sensitivity inversion using only long-term surface sites indeed shows less interannual variability of posterior OH factors, with lower error correlation between the optimization of methane emissions and OH ($r=-0.37$). We have rephrased the text to avoid misleading information "We also find that the in-situ-only inversion yields a larger interannual variability of posterior OH concentrations and thus methane lifetime than the GOSAT-only inversion (Fig.7b and Fig.S4). This is because the number and location of the observations varies from year to year, particularly for aircraft campaigns and ship cruises."**

**Comment [1-2]:** Then a comment to the added figures: Please replace figure 7b with the interhemispheric scaling factors (like Maassakkers et. al. 2019 Figure 7d, but with two values instead of one), so the size of the scaling is clear. It is not straightforward to deduce these (especially the interhemispheric ratio) based on the lifetime alone, which is why I requested this figure in the first review.

**Response [1-2]: We agree that we should show the scaling factor for OH. However, we think it**

makes more sense to show the posterior methane lifetime in Fig.7b as it is more relevant to methane budget estimates. We therefore add a new Figure.S4 to show the hemispheric OH scaling factors from the inversions. Our OH scaling factors are larger than those in Maasakkers et al. (2019) mainly because we assume a large error on prior OH fields (10% vs 3%).

[Figure]

**Figure S4.** OH scaling factors for the Southern Hemisphere (SH) and the Northern Hemisphere (NH) from the three inversions.

**Comment [1-3]:**   About Figure S2: Thank you for including this information! However the figure is quite difficult to see. The points are so tiny, I can hardly see the colours. Would it be possible to bin the data somewhat so that it's easier to interpret? A suggestion would be e.g. a few degrees of latitude, 1-km altitude bins, and then perhaps have slightly larger points. Perhaps the colour could show the mean value in the bin, and the size of the point the standard deviation within the bin? This way the noise in the NH mid-latitudes would be easier to interpret. This is just an example: I am sure there are different ways that the information could be plotted to make it clearer.
**Response [1-3]: Thank you for pointing it out. We have revised accordingly.**

[Figure]

**Figure S2.** Differences between simulated and observed aircraft methane concentrations from the GLOBALVIEWplus ObsPack data product using GEOS-Chem

with prior estimates and with posterior estimates from the in-situ-only, GOSAT-only, and GOSAT + in situ inversions. The size of the dots represents the standard deviation (SD).